# Comparison of the Green-to-desert Sahara transitions between the Holocene and the Last Interglacial

Huan Li[1,2], Hans Renssen[2,3], Didier M. Roche[2,4]

[1]School of Geographic Science, Nantong University, Tongjingdadao 999, Nantong 226007, China

[2]Vrije Universiteit Amsterdam, Faculty of Science, Cluster Earth and Climate, de Boelelaan 1085, Amsterdam, the Netherlands

[3]Department of Natural Sciences and Environmental Health, University of South-Eastern Norway, Gullbringvegen 36, N-3800 Bø i Telemark, Norway

[4]Laboratoire des Sciences du Climat et de l'Environnement, LSCE/IPSL, CEA-INSU-UVSQ-CNRS, Université Paris-

Saclay, Gif-sur-Yvette, France

*Correspondence to*: Hans Renssen (hans.renssen@usn.no)

**Abstract.** The desertification and vegetation feedbacks of the Green Sahara during the Last Interglacial (LIG) and the Holocene have been investigated by many studies. Yet the abruptness of climate and vegetation changes and their interactions are still under discussion. In this study, we apply an earth system model of intermediate complexity (iLOVECLIM) in

combination with two dynamical vegetation models (VECODE and LPJ-GUESS) to simulate climate-vegetation changes during the Holocene and the LIG to compare the patterns of North African vegetation evolutions and mechanisms of their feedbacks during these two interglacials. Our results confirmed the existence of the 'Green Sahara' during the early LIG, which is as an analogue to the 'Green Sahara' during the Holocene. During both interglacials, an overall consistent transition from vegetated Sahara to desert is shown in our results, but the amplitudes of these transitions vary. These simulated Sahara

vegetation transitions are nearly linearly related to the summer insolation declines at 20°N, resulting in faster declines of vegetation cover during the LIG than in the Holocene. The decline of vegetation cover peaks at 25%/ka at around 122 ka BP, while during the Holocene the steepest vegetation cover decline is 10%/ka at around 6 ka. Our results suggest net positive vegetation feedbacks to climates during the two interglacials. During the early LIG and Holocene, vegetation strengthens precipitation by a factor of 2 to 3 through the vegetation-albedo feedback when the vegetation cover is greater than 60%.

Vegetation cover decreases with declines of the incoming moisture transport by the atmosphere due to the reduced summer insolation at 20°N, weakening the summer monsoon during both interglacials. This desertification is accelerated when the positive vegetation-albedo feedback cannot offset the reduction of precipitation due to a weaker summer monsoon. The impacts of this positive vegetation feedback on precipitation decrease with decreased vegetation cover, during which the impacts of negative vegetation-evaporation feedbacks increase, accelerating the loss of soil moisture and vegetation cover. Overall, the

net positive vegetation feedback is strong during the early phases of both interglacials, but the vegetation transition is more abrupt during the LIG than during the Holocene due to the more rapid changes in summer insolation during the LIG. The main difference between the two interglacials is the rate of precipitation change, which is relatively gradual during the Holocene, leading to a more gradual vegetation transition in comparison to the LIG.

## 1 Introduction

From about 14.8 ka BP to 5.5 ka BP (thousand years before present), relatively high levels of summer insolation in the Northern Hemisphere (NH) led to an enhanced African monsoon, resulting in a humid and green Sahara (Kutzbach and Street-Perrot, 1985; Jolly et al., 1998; Prentice et al., 2000; Tuenter et al., 2007; Rachmayani et al., 2016). This period is therefore referred to as the African Humid Period (AHP, Ritchie et al., 1985), which is featured by shrub and grass covered land surface in Northern Africa where there is desert today (Hoelzmann et al., 2004; Lezine, 2017). Forced by the decline in summer insolation

in the NH, the termination of the AHP occurred at about 5.5 ka BP, featured by a rapid reduction in precipitation and a transition from the vegetated 'green' Sahara to desert (Jolly et al., 1998; deMenocal et al., 2000; Prentice et al., 2000; Levis et al., 2004; Liu et al., 2006, 2007; Renssen et al., 2006; Kröpelin et al., 2008; Notaro et al., 2008; Wang et al., 2008; Lezine et al., 2011; Claussen et al., 2013; Francus et al., 2013; Rachmayani et al., 2015). Several studies indicate considerable spatial variations in the nature of this transition (Shanahan et al. 2015; Tierney et al. 2017; Dallmeyer et al. 2020). Moreover, the transition rate

varies among data and models, and thus the character of the transition remains controversial (e.g., Pausata et al., 2016; Tierney et al., 2017). DeMenocal et al. (2000) inferred from marine sedimentary data that a dramatic desertification in North Africa occurred abruptly within centuries, while several other studies suggested a slower desertification in response to the gradual decline in summer insolation (Wang et al., 2005; Renssen et al., 2006; Kröpelin et al., 2008; Francus et al., 2013; Shanahan et al. 2015). One reason for these contrasting rates of desertification could be related to variations in the strength of vegetation

feedbacks that amplify the regional climate response to gradually declining orbital forcing. This is supported by a recent modelling study that showed that the abruptness of the green-desert transition is highly dependent on the model setup (Hopcroft and Valdes, 2021).

Understanding the termination of the AHP thus sheds light on the nonlinear response of the African monsoon to orbital forcing

and the role of vegetation feedbacks. There were several studies (e.g., Tuenter et al., 2007; Rachmayani et al., 2015, 2016) that investigated to what extent the AHP can be attributed to the long-term changes in the orbital forcing and the feedback processes, respectively. Tuenter et al. (2007) and Rachmayani et al. (2016) demonstrated that the long-term varying precession is the pacemaker of the African monsoon and thereby of precipitation over Northern Africa. However, a factor analysis by Rachmayani et al. (2015) found an amplifying effect of vegetation feedback on mid-Holocene precipitation over West Africa.

Moreover, the importance of surface albedo processes due to vegetation changes was supported by several studies (e.g., Levis et al., 2004; Liu et al., 2006, 2007; Notaro et al., 2008; Wang et al., 2008; Fischer and Jungclaus, 2011; Vamborg et al., 2011; Rachmayani et al., 2015). Still, the mechanism of the termination of the AHP remains controversial and only a few studies focused on the abruptness of this termination. We consider that the desertification/vegetation-transition is abrupt when the rate of the transition is faster than the gradual decline in orbital forcing and discuss the role of orbital forcing and vegetation

feedback on the termination of the AHP through comparisons with two sets of conceptual models (Liu et al., 2006 and Claussen et al., 1999, 2017). Liu et al. (2006) proposed two possibilities: either the abrupt vegetation transition in North Africa was

caused by a strong vegetation feedback combined with abrupt precipitation reduction (so called unstable collapse (UC)) or by a weak vegetation feedback combined with a strong low-frequency variability of gradual decline in precipitation (so called stable collapse (SC)). In the model experiments of Liu et al. (2006), the vegetation transition in Northern Africa was caused by SC. In another modelling study (Claussen et al., 1999), support for the first possibility (UC) was found, with vegetation collapse as a result of strong positive vegetation feedback combined with abrupt precipitation changes. In order to understand the rate of vegetation change during the termination of the AHP, the magnitude of vegetation feedbacks is thus the key factor under the gradual change in summer insolation in NH during the Holocene.

The magnitudes of climate-vegetation feedbacks have been investigated by several studies. For example, a positive vegetation-albedo feedback was found to be an important factor amplifying changes in the North African precipitation during the Holocene (Charney, 1975; Claussen and Gayler, 1997; Texier et al., 1997; Knorr and Schnitzler, 2006). Compared to desert, the vegetated surface is characterized by a lower albedo (e.g., 0.20 v.s. 0.33 in our iLOVECLIM model), which enhances the absorption of shortwave radiative energy and therefore leads to a warmer surface. Model studies suggest that vegetated surfaces in North Africa are warmer than desert, inducing larger land-sea temperature contrasts and a stronger African summer monsoon, which promotes vegetation development because of the enhanced precipitation. Therefore, the key factors adjusting the magnitude of this vegetation-albedo feedback are vegetation cover and differences in surface albedo between vegetated and bare soil surface. However, the vegetated surface affects climate not only through a lower surface albedo compared to bare soil, but also through its impacts on the evaporation and transpiration. Compared to desert, on the one hand, vegetated surfaces promote evapotranspiration, leading to evaporative cooling of the surface, resulting in lower temperature and a reduced tendency for convective precipitation. On the other hand, an increased latent heat flux (evapotranspiration) leads to more humidity in the air, which can lead to enhanced precipitation through local recirculation. Whether the net vegetation feedback through changes in evaporation and transpiration is positive or negative to precipitation is related to several factors, e.g., soil moisture availability, surface albedo, the percentage of organic matter, the amount of evaporation from different type of land surfaces, etc. For example, a dark wet soil (Levis et al., 2004) and the inclusion of canopy transpiration (Rachmayani et al., 2015) positively impact precipitation, while the feedback negatively affects precipitation when the evaporation from bare soil is stronger than transpiration from vegetated surface (Liu et al., 2007; Wang et al., 2008; Notaro et al., 2008). The net effect on the surface temperature of vegetation depends on the balance between, on the one hand, the impact of the relatively low albedo, promoting shortwave radiation absorption and warming, and on the other hand, the effect of an enhanced latent heat flux, leading to evaporative cooling.

The last interglacial (LIG) is an interesting test period with insolation changes between 129 and 120 ka BP that were stronger than in the Holocene (Fig. 1; Berger, 1978), but with similar greenhouse gas (GHG) concentrations and continental configurations (Otto-Bliesner et al., 2017). In addition, proxy-based evidence indicates that the Sahara was also green and humid in the early LIG and experienced a transition from green to desert during the LIG. This evidence for a humid period

during the LIG is based on marine sediment records (Rohling et al., 2002; Osborne et al., 2008), speleothem records (Fleitmann et al., 2011) and terrestrial sediment data (Ehrmann et al., 2017; Scussolini et al., 2019). The impacts of vegetation feedback on climate during the LIG were also studied by several modelling studies (e.g., Gröger et al., 2007; Schurgers et al., 2007). For example, Schurgers et al. (2007) suggested as much as 20% local albedo changes in Northwestern Africa during the early

LIG and more than 10% increases in global latent heat loss compared to present day. Gröger et al. (2007) simulated a more than doubling of precipitation in the Sahara (North Africa) as a result of vegetation feedbacks during the LIG. In a previous study, we have performed a series of model experiments assessing the vegetation impacts on the LIG climate (Li et al., 2020). Our global results generally compared favourably to proxy-based reconstructions, including a clear warming over North Africa (Turney and Jones, 2010; McKay et al., 2011; Li et al., 2020). In addition, we found consistent magnitudes of vegetation

feedback with previous modelling studies, including about twice as large amounts of precipitation (~60 cm/yr) and higher surface temperatures by about 2.5°C in North Africa in response to dynamical vegetation evolution during the early LIG compared to simulation with fixed pre-industrial vegetation.

The controversial abruptness of vegetation transitions and vegetation feedback during the AHP could be related to the gradual

changes in external orbital forcing during the Holocene, which may not be strong enough to counterbalance the effects of internal vegetation-climate interactions. During the LIG, conditions were different from the Holocene, and therefore, a comparison of North African vegetation transition between the LIG and the Holocene could help us disentangle the relative contribution of the external and internal processes to North African desertification.

In this paper, we first compare the LIG and Holocene transient simulations from an intermediate complexity model (iLOVECLIM) including different vegetation components with different complexities (VECODE and LPJ-GUESS). The LIG results cover 127 ka BP to 116 ka BP and are the same as reported in our previous study (Li et al.,2020). The Holocene simulations cover 8.5 ka BP to 1 ka BP. The simulations of the two interglacials with the same climate model offer the opportunity to analyze the impacts of the external forcings without model-dependence. In addition, the use of two vegetation

models increases our insight into the model-dependency of the simulated vegetation states. Thus, by performing transient climate model experiments on two different interglacials, and by applying two different vegetation models of different complexity, we are able to provide a thorough evaluation of the abruptness and the most important factors contributing to the green-desert transitions. In our simulations, VECODE was synchronously coupled, while LPJ-GUESS was asynchronously coupled to iLOVECLIM. We then assess the North African vegetation feedback by switching on and off interactive dynamic

vegetation in each coupled version. Finally, we compare the climate and vegetation simulations with the same standardized spatial scale between both interglacial periods. We analyze the vegetation transitions during both interglacials with the aim to address the following questions:

1) Are the evolutions of North African (NA) vegetation during the two interglacials simulated by iLOVECLIM with different vegetation components consistent?

2) Are the evolutions of NA vegetation different during the Holocene and LIG, and what are the dominating factors?

3) How do the NA vegetation geophysical feedbacks affect the climate during the Holocene and LIG?

## 2 Methods

### 2.1 Model descriptions

We applied the climate model iLOVECLIM version 1.0, which is an updated version of LOVECLIM 1.2 (Goosse et al., 2010;
Roche et al., 2014). This intermediate complexity model includes the main climatic system components and has been successfully applied to analyze climate-vegetation interactions during the Holocene AHP (Renssen et al., 2003; 2006) and the LIG (e.g., Li et al., 2020). In the present study, we make use of three dynamical components in iLOVECLIM representing the atmosphere (ECBilt), the oceans (CLIO), and land-based vegetation (either VECODE or LPJ-GUESS). ECBilt is a three-level quasi-geostrophic atmospheric model at T21 resolution, corresponding to ~625 km at the equator (Opsteegh et al., 1998).
ECBilt calculates the climatic forcing for the applied vegetation component (either VECODE or LPJ-GUESS). Cloud cover is prescribed according to present-day climatology (ISCCP D2 dataset, Rossow et al., 1996). In ECBilt, the hydrological cycle is closed by applying a bucket for soil moisture (Goosse et al., 2010). Every bucket is connected and drains to a nearby oceanic grid cell to calculate river runoff. The surface albedo is a function of the fraction of the grid box covered by the different surface types, i.e., ocean, sea ice, trees, desert and grass (Goosse et al., 2010). CLIO consists of a three-dimensional, free
surface ocean general circulation model coupled to a dynamic-thermodynamic sea-ice model (Goosse et al., 2010). This oceanic component has a horizontal resolution of 3° latitude by 3° longitude, and 20 unevenly spaced vertical layers in the ocean.

As in Li et al. (2020), we apply here three model configurations: (1) ECBilt-CLIO fully coupled to VECODE (ECBilt-CLIO-
VECODE); (2) ECBilt-CLIO asynchronously-coupled to LPJ-GUESS (ECBilt-CLIO_LPJ-GUESS); and (3) ECBilt-CLIO with prescribed pre-industrial vegetation (ECBilt-CLIO-LUH2). The difference among these three model configurations is their vegetation component and thereby the surface albedo and soil moisture.

In ECBilt-CLIO-VECODE, with climate forcing from ECBilt, VECODE calculates covers of two Plant Functional Types
(PFTs, grasses and trees), and bare soil as a dummy type, at an annual time step (Brovkin et al., 1997). The climate forcing for VECODE includes annual mean temperature, precipitation and GDD0 (growing degree days above 0°C). We note that using annual mean climate input implies that the vegetation in VECODE is not directly affected by seasonal variations in precipitation and temperature, although some seasonal information is provided through the GDD0. This simplification could potentially lead to an underestimation of the impact of orbital forcing on vegetation in North Africa.

Similar to ECBilt-CLIO-VECODE, LPJ-GUESS also calculates PFT cover in ECBilt-CLIO_LPJ-GUESS based on climate input from ECBilt, but LPJ-GUESS requires more specific climate input and calculates vegetation dynamics in more detail than VECODE. We apply version 3.1 of LPJ-GUESS (Smith et al., 2001; Smith et al., 2014). The climate input required by LPJ-GUESS includes monthly mean surface temperature, precipitation, and cloud cover. The model employs biophysical and
physiological process parameterizations identical to the equilibrium model BIOME3 (Haxeltine and Prentice, 1996). Vegetation dynamics in LPJ-GUESS result from growth and competitions for light, space and soil resources among 11 PFTs (Smith et al., 2001). These 11 PFTs are: Boreal needle-leaved evergreen trees, Boreal needle-leaved evergreen shade-intolerant trees, Boreal needle-leaved summer-green trees, Temperate broadleaved summer-green trees, Boreal-temperate broadleaved summer-green trees, Temperate broadleaved evergreen trees, Tropical broadleaved evergreen trees, Tropical broadleaved
evergreen shade-intolerant trees, Temperate broadleaved raingreen trees, C3-grass, and C4-grass. Physiological processes are simulated with a daily time step. The net primary production (NPP) is accrued at the end of a simulation year (Smith et al., 2001), resulting in changes in height, diameter and biomass growth of each PFT. In ECBilt-CLIO_LPJ-GUESS, LPJ-GUESS is asynchronously coupled to iLOVECLIM and calculates covers of the 11 PFTs. This asynchronous coupling procedure is identical to the study on analyzing climate-vegetation interactions during the LIG (Li et al., 2020). Here, monthly climate
inputs from the fully coupled iLOVECLIM model (ECBilt-CLIO-VECODE) are in an initial step fed to LPJ-GUESS, which simulates vegetation distributions. Then, the resulting vegetation distributions are given back to ECBilt, as a fixed vegetation component in iLOVECLIM (ECBilt-CLIO_LPJ-GUESS) during the next round of climate simulation. A new climatology from this integration is simulated and used subsequently off-line as climate forcing by LPJ-GUESS to produce a new global vegetation distribution that is subsequently used as a boundary condition by iLOVECLIM, and so on. A frequency of
vegetation simulation for 10 years is chosen, and four iterations are performed for the asynchronous couplings with this frequency. We treated vegetation results from the last iteration as vegetation responses to climate and analyze vegetation feedbacks to climate. As explained in Li et al. (2019a), the 11 PFTs of LPJ-GUESS were converted to the 3 vegetation types (trees, grasses and bare soil) used in iLOVECLIM.

In the configuration with prescribed pre-industrial vegetation (ECBilt-CLIO-LUH2), pre-industrial vegetation from the CMIP LUH2 (2012) dataset is upscaled to the same resolution as ECBilt. The LUH2 (Land-Use Harmonization, Phase 2) project prepared a harmonized set of land-use scenarios that smoothly connects the historical reconstructions of land-use with the future projections in the format required for Earth System Models. We applied the land-use data from the CMIP LUH2 dataset, at 850 AD (Fig. S1). The land surface albedo is based on this prescribed vegetation and therefore is fixed at a constant value
in ECBilt during the simulations.

In the three applied configurations, the land surface albedo and soil hydrology are computed in a simple land-surface and bucket model (LBM) embedded in ECBilt, in which land surface albedo and the maximum water volume of the bucket are a function of vegetation cover. The land surface albedo depends on the percentage of PFT cover in the LBM, which is negatively

related to tree cover. In contrast, the maximum water of the bucket is positively related to total vegetation cover. In our experiments, the fractional surface albedo of trees, grassland and desert are seasonally fixed at a constant value in the LBM, which is at 0.13, 0.20 and 0.33, respectively. The land surface albedo is thus calculated in the LBM based on dynamical vegetation cover in either VECODE or LPJ-GUESS and is then given back to ECBilt in both the fully and asynchronously coupled versions of iLOVECLIM. In ECBilt-CLIO-LUH2, however, the land surface albedo is fixed to prescribed pre-industrial level.

In both ECBilt-CLIO-VECODE and ECBilt-CLIO_LPJ-GUESS, soil hydrology depends on dynamic vegetation modelled in either VECODE or LPJ-GUESS, while soil hydrology depends on prescribed pre-industrial vegetation in ECBilt-CLIO-LUH2. The water exchanges calculated in the full hydrology model in LPJ-GUESS only impact vegetation within LPJ-GUESS, but we keep the soil hydrology calculated in the LBM in the asynchronous coupling of ECBilt-CLIO to LPJ-GUESS. In other words, the hydrology model in LPJ-GUESS affects the vegetation dynamics but the calculation of vegetation feedbacks to soil hydrology is outside of LPJ-GUESS, which is in the LBM embedded in ECBilt in the asynchronously coupled configuration (ECBilt-CLIO_LPJ-GUESS).

## 2.2 Experimental design

The LIG transient experiments are from Li et al. (2020). Our Holocene experiments were set-up in a similar way as the LIG experiments described in detail by Li et al. (2020). We performed transient simulations (Table 1) with two versions of our model configuration (ECBilt-CLIO-VECODE and ECBilt-CLIO-LUH2) for the Holocene (8.5 ka BP - 1 ka BP). The Holocene transient experiments start from equilibrium simulation of 8.5 ka BP by Li et al. (2019b). The name of each experiment reflects both the period of interest and vegetation component that were used. For example, HOL_VEC and HOL_FIX refer to transient experiments during the Holocene with vegetation component of VECODE and fixed pre-industrial vegetation, respectively. In addition, we performed a series of time-slice simulations (HOL_LPJ) with the asynchronously coupled version (ECBilt-CLIO_LPJ-GUESS) for the Holocene (8 ka BP - 1 ka BP) at 1ka intervals. These simulations were run for 1000 model years per time slice, of which we used the last 30 years for the analysis. The initial climatic forcing for each time-slice simulation is a 100-year monthly mean climatology (surface temperature, precipitation and cloud cover) from fully coupled transient simulations (HOL_VEC), and then the asynchronous coupling starts with vegetation from LPJ-GUESS. Although these time-slice simulations do not provide a continuous view of the transitions, the results at 1 ka intervals cover the full evolution from green to desert in North Africa during both the LIG and the Holocene.

The transient forcings for experiments during the LIG and the Holocene include variations in orbital parameters and greenhouse gases (GHG) concentrations. The orbital forcings for the Holocene are identical to the PMIP4 protocol (Otto-Bliesner et al., 2017). Changes in insolation received by the Earth are calculated in iLOVECLIM during transient simulations according to Berger (1978). From 8.5 ka BP to about 1 ka BP, the NH received more insolation during boreal summer and less insolation

during boreal fall and winter compared to preindustrial values. Both the Holocene and the LIG are characterized by substantial changes in the spatial and temporal distribution of insolation. The magnitude of the climatic precession changes during the LIG are stronger than the Holocene. The mean levels of eccentricity are also higher during the LIG, while the obliquity is higher during the Holocene. These differences result in higher summer insolation at 20°N during the early periods of the LIG than during the Holocene and stronger changes in the annual, seasonal and latitudinal insolation during the LIG than the Holocene (Fig. 1a, e).

The GHG concentrations during both interglacials were similar to those of the pre-industrial period. The greenhouse gases (GHG) concentrations for the Holocene in our simulations are taken from the PMIP3 protocol (https://pmip3.lsce.ipsl.fr) to keep the GHG forcing consistent with the 8.5 ka BP equilibrium simulation (8.5ka_VEC) from which the initial conditions were derived. In contrast, the GHG for the LIG is identical to the PMIP4 protocol (Otto-Bliesner et al., 2017). Compared to PMIP3, the GHG concentration has been updated in PMIP4, using new data and a revised chronology that provides a consistent history of the evolution of these gases across the Holocene (Otto-Bliesner et al., 2017). The difference in GHG forcing between PMIP4 and PMIP3 was estimated to be −0.8 W/m2 by Otto-Bliesner et al. (2017), inducing a mean global cooling by 0.24±0.04°C. A linear interpolation was applied to GHG data to get annual values for transient simulations and 100-year mean values of these interpolated values were used for asynchronously coupled simulations, e.g., mean values from 7.95 ka BP to 8.05 ka BP for the simulation HOL_LPJ in 8 ka BP.

In our experiments we kept the other boundary conditions identical to the preindustrial set up. These include the configurations of the ice sheets, the land-sea distribution, and the solar constant.

## 3 Results and discussion

### 3.1 North African climate changes

#### 3.1.1 The LIG

All the LIG experiments simulate relatively large amounts of precipitation in North Africa during the early LIG (Fig. 1b), corresponding to the relatively high summer insolation at 20°N (Fig. 1a). Background information is provided in supplementary Figures S2 to S6. During the early LIG (about 127 ka BP - 125 ka BP), LIG_VEC and LIG_LPJ simulate about twice as large amounts of precipitation (58 cm/yr) as LIG_FIX (28 cm/yr). The two simulations with dynamical vegetation (LIG_VEC and LIG_LPJ) thus indicate comparable stronger declines in precipitation than the simulation with fixed pre-industrial vegetation (LIG_FIX). From 125 ka BP to 117 ka BP, precipitation decreases by 48 cm/yr from about 58 cm/yr to less than 10 cm/yr in North Africa in LIG_VEC/LIG_LPJ, while this decline in LIG_FIX is only about 15 cm/yr.

A relatively cooler early LIG (29.0℃) in North Africa compared to pre-industrial (30.2℃) is simulated in LIG_FIX and an upward temperature trend during the LIG is seen in this simulation (Fig. 1c), which is consistent with multi-model simulations (Lunt et al., 2013; Bakker et al., 2013; Bakker et al., 2014) and PMIP4 experiments for the early LIG during which North Africa was cooler by about 1.0℃ (Williams et al., 2020). Even without the albedo effect, the precipitation in Northern Africa was still significantly higher in the early part of the interglacial due to the enhanced summer monsoon, forced by elevated insolation values. This high precipitation resulted in relatively humid soils and enhanced evaporation, leading to evaporative cooling in the first part of the LIG relative to the end of the LIG_FIX experiment. This created the positive temperature trend. Different from this positive temperature trend in LIG_FIX, the surface temperature in North Africa decreases during the LIG in LIG_VEC and LIG_LPJ. A warmer early LIG (about 31.5℃) than pre-industrial is seen in experiments LIG_VEC and LIG_LPJ, in particular in the northwestern part of NA (Li et al., 2020), which confirms the positive impacts of dynamical vegetation on surface temperature. After 121 ka BP, the temperature differences between LIG_VEC/LIG_LPJ and LIG_FIX are small, indicating the end of desertification in North Africa. During the desertification phase, between about 123 and 121ka BP, the changing rates of surface temperature reach a maximum in all three LIG simulations.

### 3.1.2 The Holocene

Precipitation declines in all simulations (Fig. 1f) during the Holocene, corresponding to the reduction in July insolation at 20°N (Fig. 1e). Similar precipitation trends in North Africa during the Holocene were suggested by previous studies (Renssen et al., 2003; Schurgers et al., 2006;). Comparable to precipitation features during the LIG, precipitation also saw larger magnitudes of decline in experiments with dynamical vegetation (HOL_VEC) than in the experiment with fixed pre-industrial vegetation (HOL_FIX). Precipitation gradually decreases from 20 cm/yr to 16 cm/yr during the period (8.5 ka BP - 2 ka BP) in HOL_FIX, corresponding to the gradual reduction in summer insolation at 20°N (Fig. 1e). From 8.5 ka BP to about 7 ka BP, the amount of precipitation in North Africa remains at a relatively high level in HOL_VEC and HOL_LPJ (45 cm/yr) in comparison to the value (22 cm/yr) in HOL_FIX. Compared to this early period, the amount of precipitation decreases from 40 cm/yr to 30 cm/yr with increased variability during the period between 7 ka BP to 5 ka BP in HOL_VEC, and this precipitation trend is more gradual in HOL_FIX than in HOL_VEC. Subsequently, the annual precipitation decreases gradually to a relatively low level (around 20 cm/yr) at 2 ka BP, in particular in experiments HOL_VEC and HOL_FIX. However, the reduction in precipitation in HOL_LPJ is much more modest (from about 40 to 35 cm/yr) over the 8.5 to 2 ka BP period, which could be related to the simulated vegetation distribution and spatial divergence in North Africa (Fig. 3).

During the period from 8.5 ka BP to 7 ka BP, a warmer North Africa (31.2℃) compared to pre-industrial (30.2℃) is suggested by HOL_VEC and HOL_LPJ (Fig. 1g), but a lower surface temperature is shown in HOL_FIX (29.0℃). Compared to the temperature evolution in HOL_FIX, a subsequent cooling trend from 7 ka BP to 2 ka BP is found in HOL_VEC, indicating positive impacts of vegetation feedbacks on temperature consistent with earlier studies (Renssen et al., 2003; Bakker et al., 2014). In contrast, no clear temperature change is simulated in HOL_LPJ, similar to HOL_FIX. This difference in surface

temperature between HOL_VEC and HOL_LPJ reflects different magnitudes of vegetation feedbacks under different dynamical vegetation conditions.

**3.2 North African vegetation evolution and feedbacks**

**3.2.1 The LIG**

Vegetated North Africa experienced desertification during the LIG in experiments LIG_VEC and LIG_LPJ. Compared to LIG_FIX, the magnitudes of vegetation feedbacks to climate in both experiments vary with simulated vegetation distributions through vegetation-induced changes in albedo and surface evaporation. During the early LIG (127 ka BP -125 ka BP), the Sahara region is covered with vegetation (>70%) in both LIG_VEC and LIG_LPJ (Fig. 2a). Vegetation cover mainly declines in the northwestern part of the region from 123 ka BP onwards (Fig. 3). The decreases in vegetation cover accelerate from 123 ka BP in these two experiments (Fig. 2a), with a steeper trend in LIG_VEC. The sharp decline in vegetation in LIG_VEC is simultaneous with a strong reduction in precipitation (Fig. 1b), but also with stronger cooling (Fig. 1c), showing that feedbacks between vegetation and climate are behind the enhanced desertification. The rate of this decline reaches a peak at 122 ka BP in LIG_VEC, while it is later in LIG_LPJ (at about 121 ka BP). This faster vegetation transition in LIG_VEC than in LIG_LPJ is related to the different complexities of vegetation components, as we discussed in detail in an earlier studies (Li et al., 2019a; 2020). We found that the complexity of VECODE and LPJ-GUESS affects vegetation simulations mainly through diversity when the atmospheric $CO_2$ level is around pre-industrial level (280 ppmv), while the difference in complexity affects vegetation simulations mainly through ecophysiological processes when the atmospheric $CO_2$ level is largely different from this pre-industrial value. This finding that higher vegetation diversity leads to a more gradual vegetation change is consistent with previous studies on vegetation transitions during the Holocene (Claussen et al., 2013; Hely et al., 2014; Li et al., 2019a).

Corresponding to the vegetation transitions in the LIG experiments, the impacts of vegetation-climate feedbacks are revealed by comparisons between LIG_VEC/LIG_LPJ and LIG_FIX. Enhanced vegetation covers (Fig. 2a) accompanied by relatively high levels of precipitation (Fig. 1a, 1b) are simulated in LIG_VEC and LIG_LPJ, indicating the effects of vegetation feedbacks on precipitation. These effects are positive with maximum magnitudes during the early LIG due to a high vegetation cover (Fig. 2a). This positive feedback of vegetation to precipitation has been associated with changes in albedo (Charney, 1975; Eltahir, 1996; Eltahir and Gong, 1996; Braconnot et al., 1999; Schurgers et al., 2007; Levis et al., 2004; Vamborg et al., 2011). During the LIG, the effect of vegetation feedback on precipitation decreases with the decreasing vegetation cover due to the increasing surface albedo, and this effect on precipitation becomes close to 0 cm/yr at 121 ka BP after desertification.

**3.2.2 The Holocene**

Vegetation cover (Fig. 2b) decreases gradually from 60% to 5% during the period between 8.5 ka BP and 2 ka BP in HOL_VEC, while more fluctuations occur in HOL_LPJ. A minor decline in vegetation cover around 6 ka BP is shown in

HOL_LPJ, and this decline continues with large variability after 6 ka BP. Such gradual vegetation declines have also been found in terrestrial records (e.g., Kröpelin et al., 2008; Amaral et al., 2013) and simulations (e.g., Renssen et al., 2006;
Schurgers et al., 2006), but they are inconsistent with the abrupt decline suggested by dust records (de Menocal et al., 2000) and some simulation studies (Liu et al., 2007; Notaro et al., 2008). One important reason for this discrepancy is the spatial heterogeneity during the vegetation transition. For example, Renssen et al. (2003) found a gradual vegetation decline in most of the Sahara, yet Liu et al. (2007) suggested abrupt vegetation decrease in the eastern part of the Sahara. The other reason of the gradual vegetation decline and large variability in HOL_LPJ could be the asynchronously coupling of LPJ-GUESS to
iLOVECLIM. In such coupling, the simulated vegetation is averaged every 10 years and is then passed back to ECBilt as static land surface parameters. Thus, the vegetation transition could be easily smoothed. Likewise, the discrepancy between the vegetation decline simulated by LPJ-GUESS offline (directly forced by climate from HOL_VEC) and HOL_LPJ also increases from 6 ka BP, confirming the increasing impacts of the asynchronous coupling on vegetation during the Holocene when the summer insolation and precipitation decline gradually (Fig. 1). The large error bars for the HOL_LPJ snapshot experiments
for 5 and 4 ka (Fig. 2b) indicate that there was a strong increase in the interannual variability in the simulation of PFT cover. This is likely related to one or more PFTs being very close to their climatic limit, such that relatively small variations in precipitation cause large shifts between the area covered by the involved PFT. In this case, also bare ground was involved.

To investigate the spatial heterogeneity of Holocene vegetation transitions in the Sahara, we calculated regional-scale (the
western and eastern parts of North Africa) features for climate and vegetation changes in HOL_VEC and HOL_LPJ. Similar to the evolution of vegetation in the whole Sahara, the vegetation cover declines gradually during the Holocene in the western part of the Sahara (Fig. 4a), but it experiences a slight drop around 6 ka BP in the eastern part, especially in HOL_VEC (Fig. 4b). Consistently, an earlier end of the AHP in the eastern than the western part of the Sahara is seen in HOL_VEC (Fig. 4), indicating considerable variations in the AHP termination, both in space and time. This spatial and temporal complexity implies
that the termination of the AHP could have been locally abrupt. Regionally different responses to the Holocene insolation decrease are also suggested by Shanahan et al. (2015) and Dallmeyer et al. (2020, 2021). Moreover, the rate of decline at the end of the AHP indicate complexity. The gradual decline in vegetation cover over western North Africa is consistent with simulations by Renssen et al. (2006), in response to the gradual declines in precipitation and summer insolation at 20°N. In contrast, although the slight drop around 6 ka BP over eastern North Africa is also seen in simulations by Liu et al. (2007), the
magnitude of our vegetation decline is much weaker than in their study that shows a decline of about 30% from 6 ka BP to 5 ka BP. The different magnitudes could be related to the differences in model complexity and also to the selection of a specific area in eastern North Africa. This spatial heterogeneity suggests that the abruptness of vegetation decline is a regional-scale feature (Brovkin and Claussen, 2008).

The magnitudes of vegetation feedback to climate during the Holocene depend on differences in vegetation cover. The difference between HOL_VEC and HOL_FIX, interpreted here as the impact of dynamical vegetation on precipitation,

declines gradually from 27 cm/yr at 8.5 ka BP to 5 cm/yr at 2 ka BP in response to the reduced vegetation cover (Figs. 1f, 3). A similar impact of 25 cm/yr on precipitation is shown for HOL_LPJ (Fig. 1f) at 8 ka BP due to the similar level of vegetation cover to HOL_VEC. However, the trends of this vegetation feedback indicate large fluctuations in HOL_LPJ (Fig. 1f, 2b) due to its complex vegetation cover changes after 6 ka BP. The increased precipitation (roughly a factor of 2) at around 8 ka BP in HOL_LPJ and HOL_VEC compared to the fixed pre-industrial vegetation counterpart (HOL_FIX) is consistent with strong vegetation feedbacks suggested by previous simulations (Claussen and Gayler, 1997; Renssen et al., 2003; Rachmayani et al., 2015) and mid-Holocene reconstructions (Bartlein et al., 2011). The strong vegetation feedback to precipitation shows that the pure insolation forcing (as in HOL_FIX) is not sufficient to reproduce the reconstructed rainfall, implying that vegetation feedbacks substantially amplify the rainfall response in North Africa in the early Holocene.

### 3.3 Comparison between the LIG and Holocene

The climate and vegetation changes in North Africa during the LIG and the Holocene share many common aspects (Fig. 1 and 2): 1) the amount of precipitation decreases in response to the reduction in summer insolation at 20°N, 2) the land surface in this region experienced a transition from a vegetated cover to desert cover, 3) the positive impacts of vegetation feedback on precipitation reaches its maximum during the early period of both interglacials. In addition to these common features, the main differences between the two interglacials are the stronger amplitudes of climate and vegetation changes during the LIG compared to the Holocene.

Based on the results of our experiments, during the early period of both the LIG and Holocene, land surface was dominantly covered by vegetation with only a small proportion of desert in North Africa. This is in agreement with the 'Green Sahara' suggested by reconstructions (e.g., Prentice et al., 2000) and simulations (e.g., Renssen et al., 2003; Schurgers et al., 2006, 2007). The summer insolation at 20°N around 125 ka BP is higher by 15 W/m$^2$ than around 8.5 ka BP (490 W/m$^2$), resulting in a higher amount of precipitation in the LIG over North Africa by about 12 cm/yr and slightly higher vegetation cover by 14% around 125 ka BP than the early Holocene. Compared to the pre-industrial, the higher summer insolation during the early LIG and Holocene induces a stronger summer monsoon, resulting in more moisture transported from the Atlantic to the continent and therefore supporting a larger vegetation cover (Fig. 2). In addition to the amount of precipitation induced by the high insolation, vegetation feedbacks also positively affect the amount of precipitation in North Africa. The large vegetation cover amplified regional climate through changes in surface albedo and evaporation, acting as biogeophysical feedbacks to climate. The amounts of precipitation in experiments with dynamical vegetation are about twice the values in experiments with fixed pre-industrial vegetation during both interglacials when their total vegetation cover is above 60%. This fact indicates that vegetation cover plays a central role in the magnitude of vegetation feedbacks, which is consistent with the highlighted role of vegetation on climate by studies based on fully coupled Earth system models (Braconnot et al., 2012; Shanahan et al., 2015; Dallmeyer et al., 2020).

During the transition period from vegetation to desert, vegetation declined faster during the LIG (about 12%/ka) than in the Holocene (about 4%/ka), corresponding to a faster reduction in summer insolation at 20°N (compare Fig. 2 with Figs. 1a and 1e). Indeed, in the LIG between 120 and 118 ka, the summer insolation decreases to a value well below that of the Holocene. This faster transition during the LIG can also be seen in the development of precipitation, evaporation and soil moisture in all our simulations (Fig. 1). In the Sahara, the amount of precipitation is the most important limiting factor for vegetation growth.

In our simulations, the vegetated Sahara is sustained when the amount of precipitation is larger than about 40 cm/yr (Fig. 5), which is consistent with earlier studies suggesting a requirement of precipitation of more than 37 cm/yr for sustaining vegetation in the region between 18-23°N and 11-36°E (Liu et al., 2006; Hopcroft et al., 2017). During the vegetation transition in both interglacials, the reduction in vegetation cover is associated with an increasing surface albedo, leading to decreased amounts of precipitation (Fig. 5), showing the active role of surface albedo in the vegetation feedback to precipitation.

In the transition periods during both interglacials, the amount of precipitation decreases with the reduction in summer insolation at 20°N, and the slope of the relationship between precipitation and summer insolation at 20°N increases when this insolation is greater than about 470 W/m$^2$ (Fig. 6). The nearly linear relationship between precipitation and summer insolation at 20°N indicates that insolation is the driver of precipitation change. Moreover, the enhanced slopes in the experiments with dynamical

vegetation (green and blue circles in Fig. 6) suggest that the positive biogeophysical vegetation feedbacks adjust the amount of precipitation in North Africa in response to the high vegetation cover (Fig. 5a, 5d). In contrast, several studies found negative vegetation feedbacks to precipitation during the Holocene through the mechanism that evaporation from the bare ground appears to be stronger than evapotranspiration from grassland under wet conditions, during which the differences in surface albedo between wet soil and vegetation are small (Liu et al., 2007; Notaro et al., 2008; Wang et al., 2008). However, this

negative feedback is not seen in our Holocene experiments, which could be related to the fixed surface albedo of bare soil that is not a function of water content in both our dynamical vegetation experiments, leading to an overestimation of the positive vegetation-albedo feedback. In addition, no seasonal variations in albedo were applied, which likely also affected the strength of this feedback.

In the LIG the simulated precipitation was lower than in the Holocene, even with similar insolation values at 20°N (Fig. 6). This is likely related to differences in the thermal contrast between the Atlantic Ocean and the continent, and to differences in the strength of vegetation feedbacks. The thermal contrast determines the strength of the monsoonal precipitation, and since the Atlantic Ocean was slightly warmer in the LIG than in the Holocene, this could partly explain a stronger precipitation in the Holocene with the same summer insolation. In addition, comparing 122 ka with 8 ka (Fig. 3), the vegetation cover is more

extensive in the early Holocene, which enforces the precipitation through the vegetation-climate feedbacks.

Although the positive vegetation feedbacks enhance the amount of precipitation in North Africa during both interglacials, the mechanism of vegetation transitions during the two interglacials could be different, in particular during the transition periods.

During the transition in the LIG (Fig. 1b, 2a), the decline in vegetation and precipitation are characterized by large variability, but these reductions during the Holocene are relatively gradual (Fig. 1f, 2b). Based on a series of sensitivity experiments by Liu et al. (2006), vegetation collapse in North Africa could occur either when the vegetation feedback to precipitation is relatively strong with rapid declines in precipitation or when the feedback is weak with gradual decreasing precipitation. In their study, the vegetation feedback to precipitation is strong when the amount of vegetation-induced precipitation is greater than the lowest requirement of precipitation for a vegetated Sahara. In our case, the vegetation feedbacks to precipitation are greater than the lowest requirement of precipitation for vegetated Sahara during the early periods of both interglacials. Therefore, the abrupt vegetation transition during the LIG is likely to be due to the combination of the strong vegetation feedback and rapidly decreasing precipitation, while the gradual vegetation transition during the Holocene is a result of the combination of strong vegetation feedback with gradually decreasing precipitation. Moreover, during both interglacials, larger variability in precipitation and vegetation is seen in experiments with dynamic vegetation in VECODE than in LPJ-GUESS. One reason of this difference is the higher complexity of LPJ-GUESS, which simulates more gradual vegetation responses when more plant types interact with climate, consistent with other studies (Claussen et al., 2013; Hely et al., 2014; Li et al., 2019a).

## 3.4 Uncertainties

One source of uncertainty is related to the spatial vegetation heterogeneity, in particular during the Holocene when the reduction in summer insolation at 20°N is rather gradual compared to the LIG. In order to find the signal of abrupt vegetation decline in Sahara during the Holocene, we investigated the evolutions of vegetation and climate over the western and eastern part of North Africa. We do find consistent vegetation declines with previous studies and regional-scale features of vegetation changes. However, the coarse resolution of our climate model limits our scope of vegetation heterogeneity into details, which smooths the simulated results and affects the abruptness of transitions especially when the external forcing (insolation) is around the threshold of vegetation transition. In addition, Bathiany et al. (2012) pointed out the importance of spatial interactions, i.e., abrupt changes in one region can be induced by critical transitions in the neighboring region. These abrupt changes can occur in one region triggered by strong vegetation-atmosphere interaction in a neighboring region, which is called 'inducing tipping of ecosystems' by Bathiany et al. (2013a; 2013b). However, these 'inducing tipping of ecosystems' are not seen in our simulations. We therefore recommend more studies on regional-scale vegetation-climate interactions by fully coupled AOVGCMs, as well as comparisons with results from EMICs to explore the regional-scale features and to improve our understanding of climate and vegetation sensitivity to these small-scale features in North Africa. In addition, these model results should be thoroughly evaluated using proxy-based reconstructions of the green-desert transition during the Holocene and LIG.

A second major source of uncertainty is related to the land surface parameters (surface albedo and vegetation covers) provided to ECBilt, resulting in vegetation feedbacks to climate. In our study, the value of albedo for each PFT is prescribed, implying

that this value in each grid cell is decided by fractions of PFTs and is passed back to ECBilt through full- or asynchronous-coupling. The prescribed albedo value for bare soil (0.33) could exaggerate the simulated positive vegetation-albedo feedback during the interglacials, especially during the transition periods when the soil moisture could be higher than typically found in
deserts. This overestimation is obvious, in particular during the Holocene when the summer insolation at 20°N declines rather gradually. Therefore, more studies on albedo values of bare soil with different water retentions and the parameterizations of these mechanisms in climate models could improve the understanding of vegetation-climate interactions.

In addition, there are two vegetation models in this study. VECODE, fully coupled to ECBilt, suggests more abrupt vegetation
and climate changes than LPJ-GUESS which is asynchronously coupled to ECBilt during both interglacials. The main reason for this difference in our results could be related to their different complexity. The faster vegetation transition in VECODE confirms the hypothesis that the low diversity implies a high likelihood for abrupt transitions (Scherer-Lorenzen, 2005; Claussen et al., 2013; Hely et al., 2014; Li et al., 2019a). Moreover, the simulated vegetation in the asynchronously coupled LPJ-GUESS is averaged every 10 years and is then passed back to ECBilt as static land surface parameters. Hence, the
transition from green Sahara to desert is easily smoothed, and the feedback is thereby underestimated. In order to narrow down the uncertainty in terms of the asynchronous coupling during simulations, we prefer the full dynamical coupling of LPJ-GUESS to ECBilt in the future, which could enhance the robustness in climate-vegetation interaction simulations.

## 4. Conclusions

In this study, we compare the climate and vegetation changes over North Africa during the Holocene and the LIG using the
climate model iLOVECLIM in which the vegetation component is either the fully coupled VECODE model or the asynchronously coupled LPJ-GUESS model. We address three scientific questions: 1) Are the evolutions of NA vegetation during the two interglacials simulated by iLOVECLIM with different vegetation components consistent? 2) Are the evolutions of NA vegetation different during the Holocene and LIG and what are the dominating factors? 3) How do the NA vegetation geophysical feedbacks affect the climate during the Holocene and LIG?


During both interglacials, we find overall agreement with a transition from vegetated Sahara (with more than about 60% vegetation cover) to desert and positive vegetation feedback to climate. The simulations with a relatively complex vegetation component (LPJ-GUESS) suggest a gradual vegetation transition compared to simulations with a simple vegetation component (VECODE). This more gradual vegetation transition in simulations with LPJ-GUESS is a result of a smoothness of the
desertification trend due to the increasing vegetation diversity and features of different PFTs in the more complex vegetation component. Except for the impacts of the applied vegetation model, the simulated Sahara vegetation transitions are nearly linearly related to the decreasing summer insolation at 20°N. Due to the faster decrease of 20°N summer insolation during the

LIG than in the Holocene, the vegetation cover experienced a stronger reduction in the LIG. The decline of vegetation cover peaks at 25%/ka at around 122 ka BP, while the steepest vegetation cover change is 10%/ka at around 6 ka.


In both interglacials, the abruptness of vegetation changes is nearly linearly related to the changes of summer insolation at 20°N, and the effects of vegetation feedback to climate positively depend on vegetation cover. Vegetation feeds back to precipitation by a factor of 2 to 3 when the vegetation cover is greater than 60% during the early LIG and Holocene through vegetation-albedo feedbacks. Due to the different magnitudes of declines in summer insolation at 20□N, the abruptness of

vegetation transitions during the LIG and the Holocene is different. During the LIG, the abrupt transition is a result of strong vegetation feedback combined with rapidly decreasing precipitation, while the gradual transition during the Holocene is related to the strong vegetation feedback and gradually decreasing precipitation.

**Author contribution**

The study was designed by all authors. HL performed the simulations and wrote the manuscript with contributions of DMR and HR. All authors contributed to the analysis and interpretation of the model results.

**Competing interests**

None

**Acknowledgements**

This work was supported by the China Scholarship Council, the National Natural Science Foundation of China (4210010473), the Natural Science Foundation of Jiangsu Province, China (BK20210848), the Natural Science Foundation of Nantong (JC2021161), and Jiangsu Provincial Double-Innovation Doctor Program (JSSCBS20211094). DMR is supported by CNRS/INSU and by the Vrije Universiteit Amsterdam.

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

**Table 1: Overview of the performed experiments**

| Pre-industrial experiments | |
|---|---|
| PI_VEC | Pre-industrial run with **dynamical** vegetation in **VECODE** (ECBilt-CLIO-VECODE) |
| PI_FIX | Pre-industrial run with **fixed** vegetation from **CMIP LUH2** (ECBilt-CLIO-LUH2) |
| PI_LPJ | Pre-industrial run with **asynchronously** dynamical vegetation in **LPJ-GUESS** (ECBilt-CLIO-LPJ-GUESS) |
| **LIG experiments** | |
| 127K_VEC | 127 ka BP **ctrl run** with dynamical vegetation in **VECODE** (ECBilt-CLIO-VECODE) |
| **LIG_VEC** | The LIG (127 ka BP- 116 ka BP) run with **transient** insolation forcing and dynamical vegetation in **VECODE** (ECBilt-CLIO-VECODE) |
| **LIG_FIX** | The LIG (127 ka BP- 116 ka BP) run with **transient** insolation forcing and fixed vegetation from **CMIP LUH2** (ECBilt-CLIO-LUH2) |
| **LIG_LPJ** | The LIG (125 ka BP- 117 ka BP) run with 1 ka interval insolation forcing and **asynchronously** dynamical vegetation in **LPJ-GUESS** (ECBilt-CLIO-LPJ-GUESS) |
| **Holocene experiments** | |
| 8.5ka_VEC | 8.5 ka BP ctrl run with dynamical vegetation in **VECODE** (ECBilt-CLIO-VECODE) |
| **HOL_VEC** | The Holocene (8.5 ka BP- 1 ka BP) run with **transient** insolation forcing and dynamical vegetation in **VECODE** (ECBilt-CLIO-VECODE) |
| **HOL_FIX** | The Holocene (8.5 ka BP- 1 ka BP) run with **transient** insolation forcing and fixed vegetation from **CMIP LUH2** (ECBilt-CLIO-LUH2) |
| **HOL_LPJ** | The Holocene (8 ka BP- 1 ka BP) run with 1 ka interval insolation forcing and **asynchronously** dynamical vegetation in **LPJ-GUESS** (ECBilt-CLIO-LPJ-GUESS) |

**Figure captions**

**Figure 1: Evolution of 20°N July insolation, North African precipitation, surface temperature and values of evaporation minus precipitation during the LIG (a, b, c, d) and Holocene (e, f, g, h). Our target area is delimited by 10°W to 35°E, and 15°N to 30°N. The blue lines and red lines are simulated results from transient experiments with dynamical vegetation and fixed pre-industrial vegetation, respectively. The green dots with error-bars are simulated results from experiments with asynchronous-coupled 'dynamical' vegetation.**

**Figure 2: Total vegetation cover during the LIG (a) and the Holocene (b). The blue lines are simulated results from transient experiments with dynamical vegetation (LIG_VEC and HOL_VEC). The light green dots with error-bars are simulated results from experiments with asynchronous-coupled 'dynamical' vegetation (LIG_LPJ and HOL_LPJ). The dark green dots with error-bars are vegetation results from LPJ-GUESS forced offline with climatic forcing from LIG_VEC and HOL_VEC. The blue, red and light green horizontal lines indicate pre-industrial vegetation cover from PI_VEC, PI_FIX and PI_LPJ, respectively.**

**Figure 3: Total vegetation cover anomalies during the LIG (left two columns) and Holocene (right two columns) in the top panel, compared to simulated pre-industrial vegetation (PI, bottom panels) from VECODE (left) and LPJ-GUESSE (right).**

**Figure 4: Evolution of western (a) and eastern (b) part of North African precipitation, surface temperature, evaporation, soil moisture and total vegetation cover during the Holocene. The longitudinal boundaries were between 10°W and 10°E (western part), and 10°E-35°E (eastern part), both with latitudinal limits at 15N and 30N.**

**Figure 5: The relationship between a) precipitation and vegetation cover; b) vegetation cover and surface albedo; c) surface albedo and precipitation with dynamical vegetation from VECODE during two interglacials; d) precipitation and vegetation cover; e) vegetation cover and surface albedo; f) surface albedo and precipitation with dynamical vegetation from LPJ-GUESS during two interglacials.**

**Figure 6: The relationship between insolation at 20°N and precipitation with dynamic vegetation from LPJ-GUESS (a) and**
**VECODE (b) during two interglacials, with blue dots representing the LIG and green dots representing the Holocene. For reference, the results of HOL_FIX, LIG_FIX and PI are also included.**

**Figures**

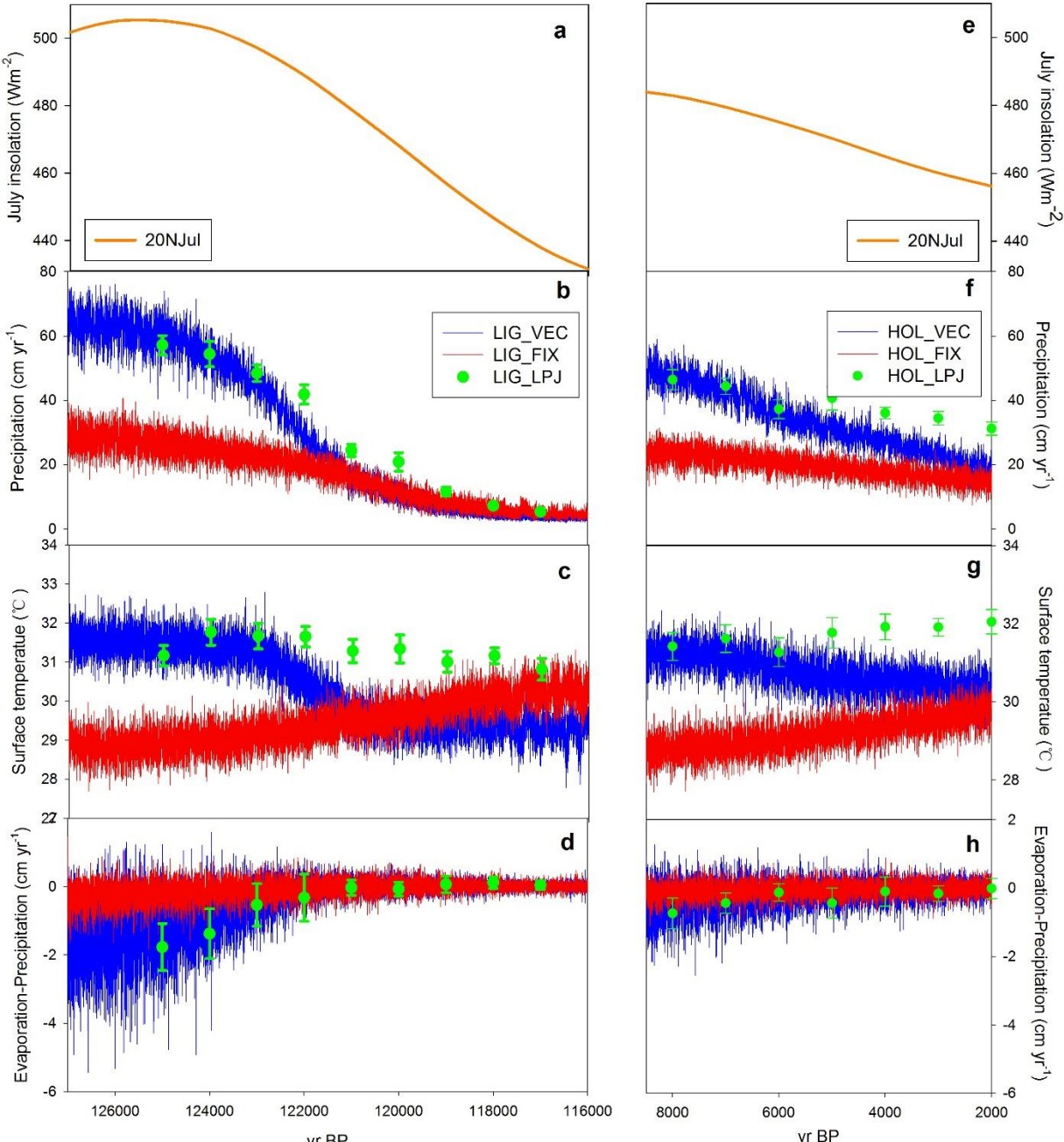

**Figure 1: Evolution of 20°N July insolation, North African precipitation, surface temperature and values of evaporation minus precipitation during the LIG (a, b, c, d) and Holocene (e, f, g, h). Our target area is delimited by 10°W to 35°E, and 15°N to 30°N. The blue lines and red lines are simulated results from transient experiments with dynamical vegetation and fixed pre-industrial vegetation, respectively. The green dots with error-bars are simulated results from experiments with asynchronous-coupled 'dynamical' vegetation.**

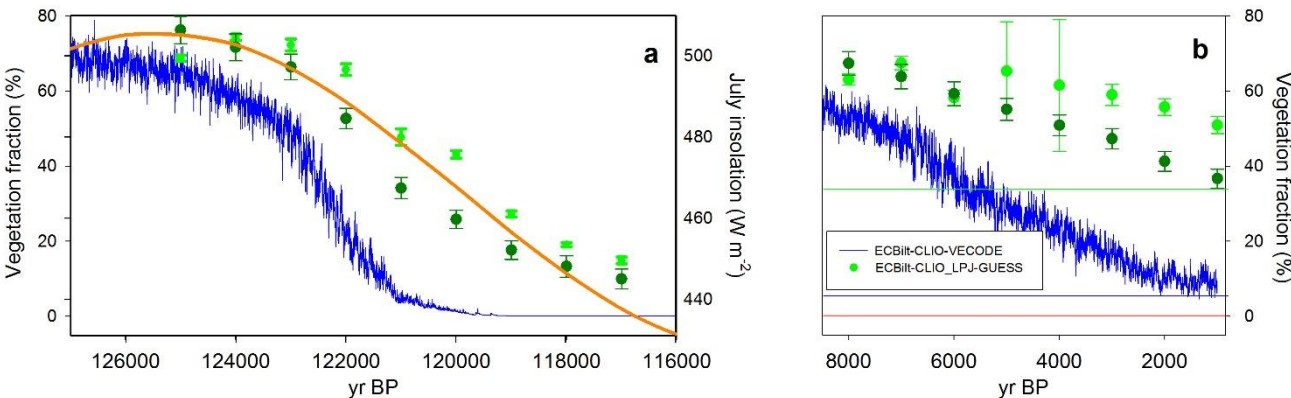

**Figure 2: Total vegetation cover during the LIG (a) and the Holocene (b). The blue lines are simulated results from transient experiments with dynamical vegetation (LIG_VEC and HOL_VEC). The light green dots with error-bars are simulated results from experiments with asynchronous-coupled 'dynamical' vegetation (LIG_LPJ and HOL_LPJ). The dark green dots with error-bars are vegetation results from LPJ-GUESS forced offline with climatic forcing from LIG_VEC and HOL_VEC. The blue, red and light green horizontal lines indicate pre-industrial vegetation cover from PI_VEC, PI_FIX and PI_LPJ, respectively.**

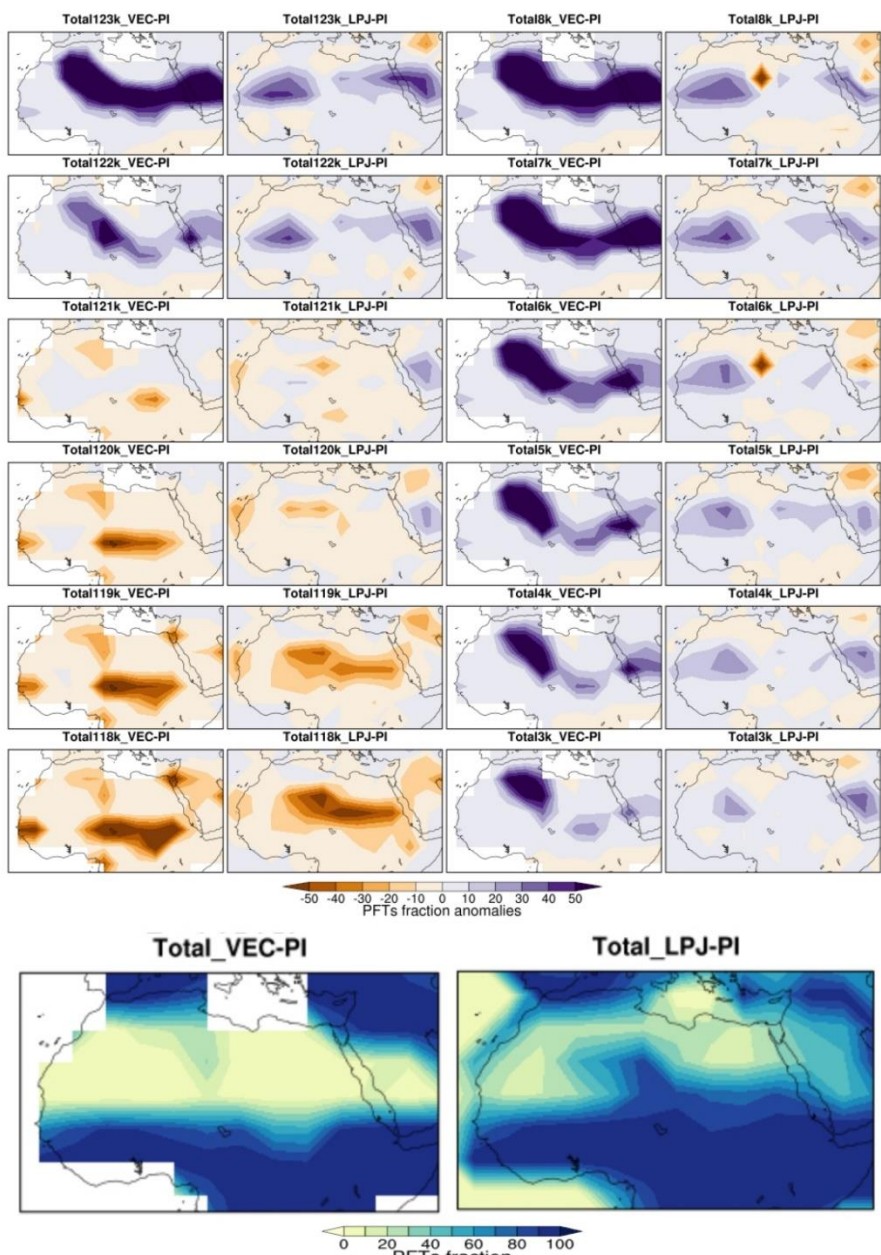

**Figure 3: Total vegetation cover anomalies during the LIG (left two columns) and Holocene (right two columns) in the top panel, compared to simulated pre-industrial vegetation (PI, bottom panels) from VECODE (left) and LPJ-GUESSE (right).**

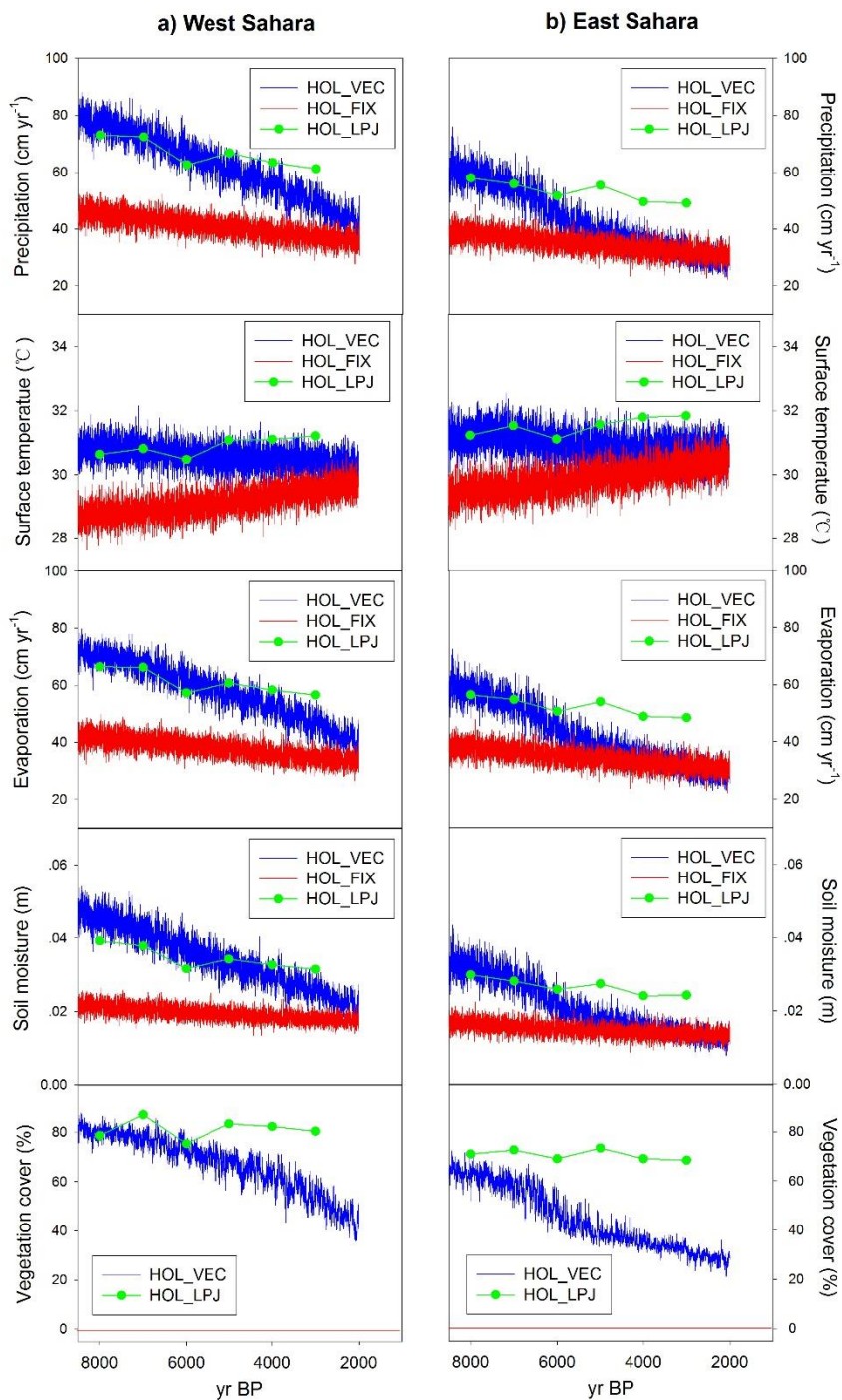

**Figure 4: Evolution of western (a) and eastern (b) part of North African precipitation, surface temperature, evaporation, soil moisture and total vegetation cover during the Holocene. The longitudinal boundaries were between 10°W and 10°E (western part), and 10°E-35°E (eastern part), both with latitudinal limits at 15N and 30N.**

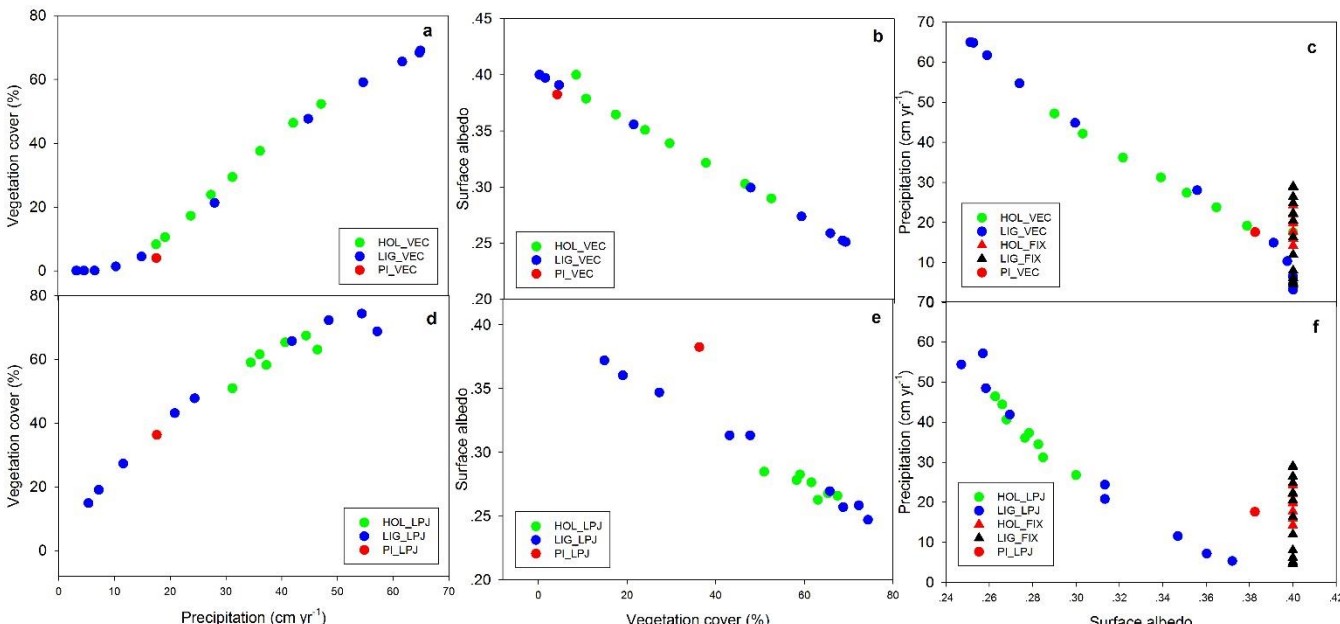

**Figure 5: The relationship between a) precipitation and vegetation cover; b) vegetation cover and surface albedo; c) surface albedo and precipitation with dynamical vegetation from VECODE during two interglacials; d) precipitation and vegetation cover; e) vegetation cover and surface albedo; f) surface albedo and precipitation with dynamical vegetation from LPJ-GUESS during two interglacials.**

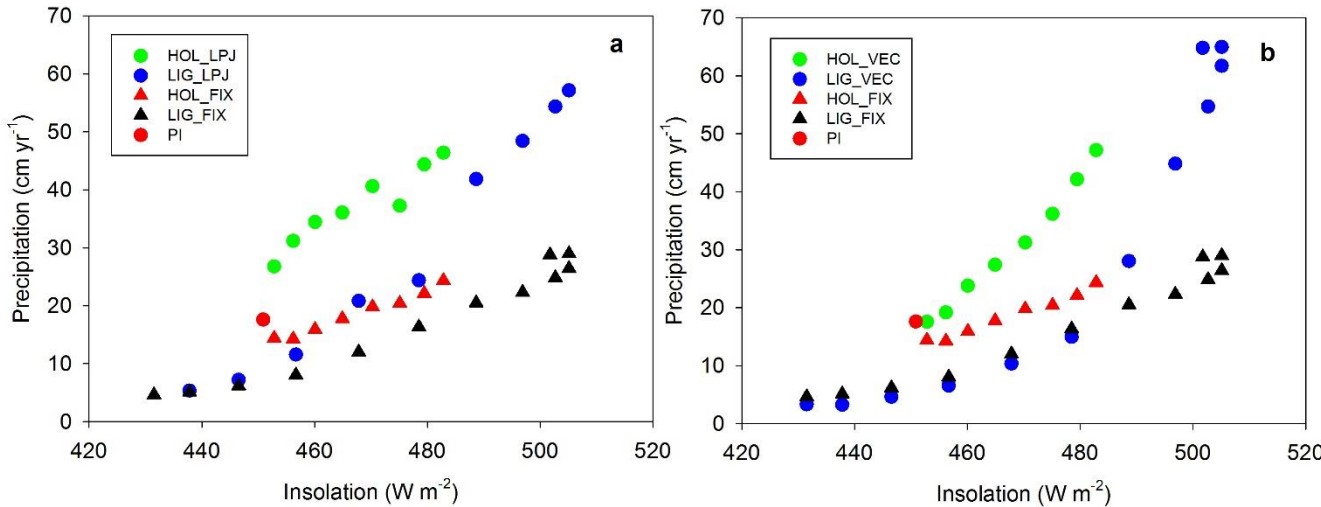

**Figure 6: The relationship between insolation at 20°N and precipitation with dynamic vegetation from LPJ-GUESS (a) and**
**VECODE (b) during two interglacials, with blue dots representing the LIG and green dots representing the Holocene. For reference,**
**the results of HOL_FIX, LIG_FIX and PI are also included.**

**Supplementary figures**

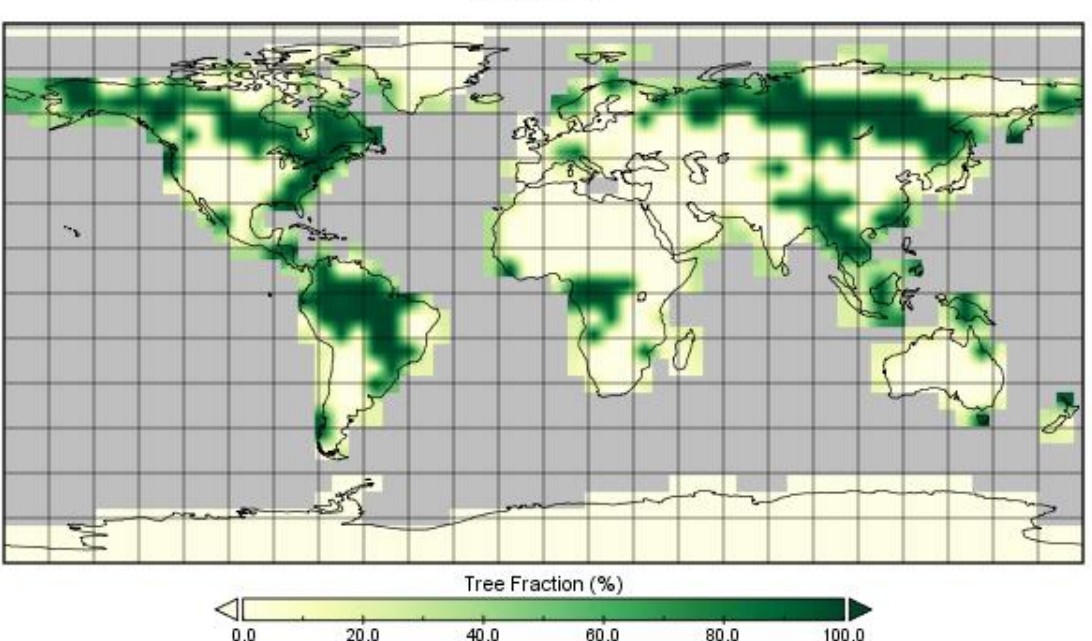

Tree Fraction

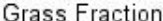

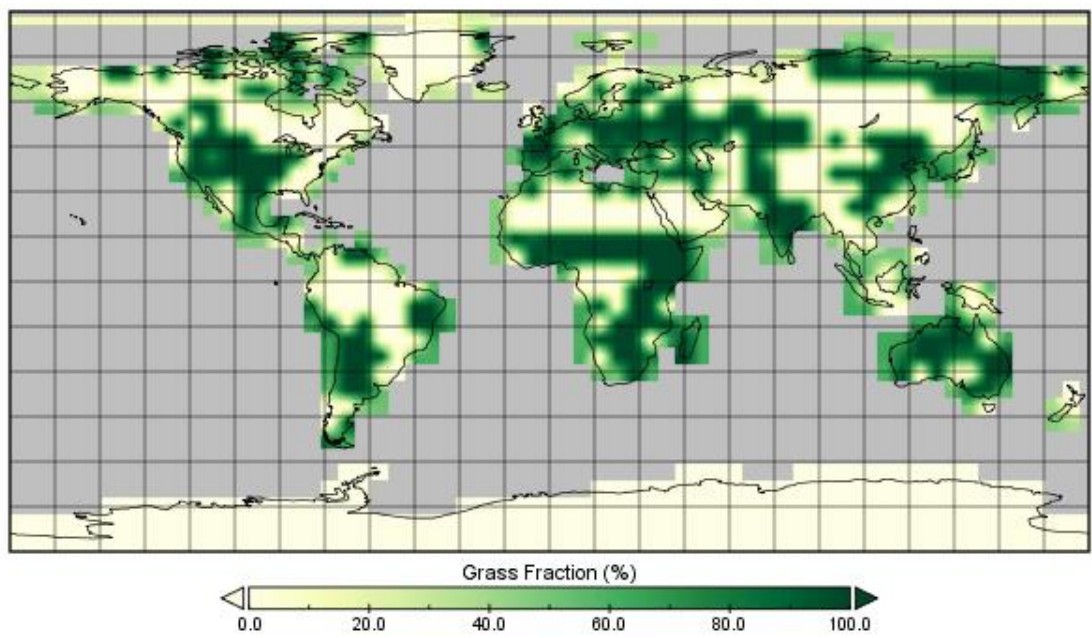


**Figure S1. Pre-industrial tree and grass cover at 850 AD from LUH2 dataset**

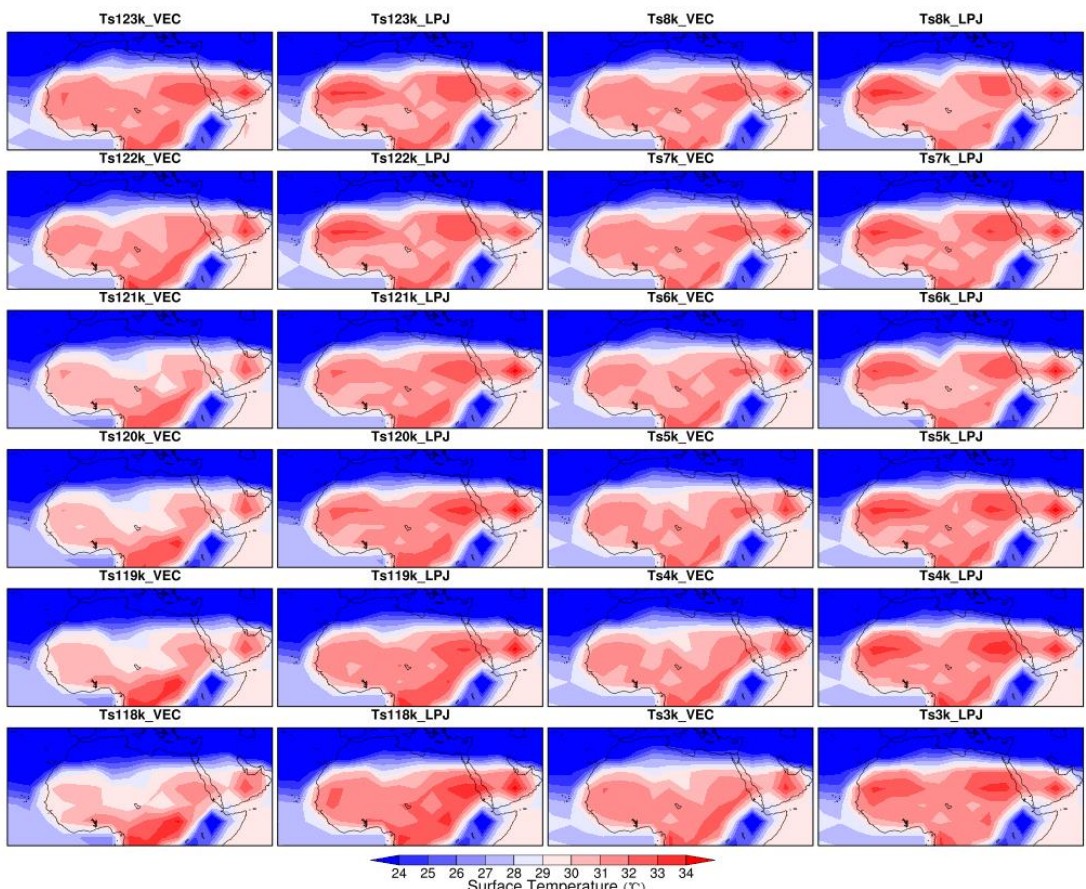

**Figure S2. Surface temperature during the LIG (left two columns) and Holocene (right two columns) from VECODE (left) and LPJ-GUESS (right).**

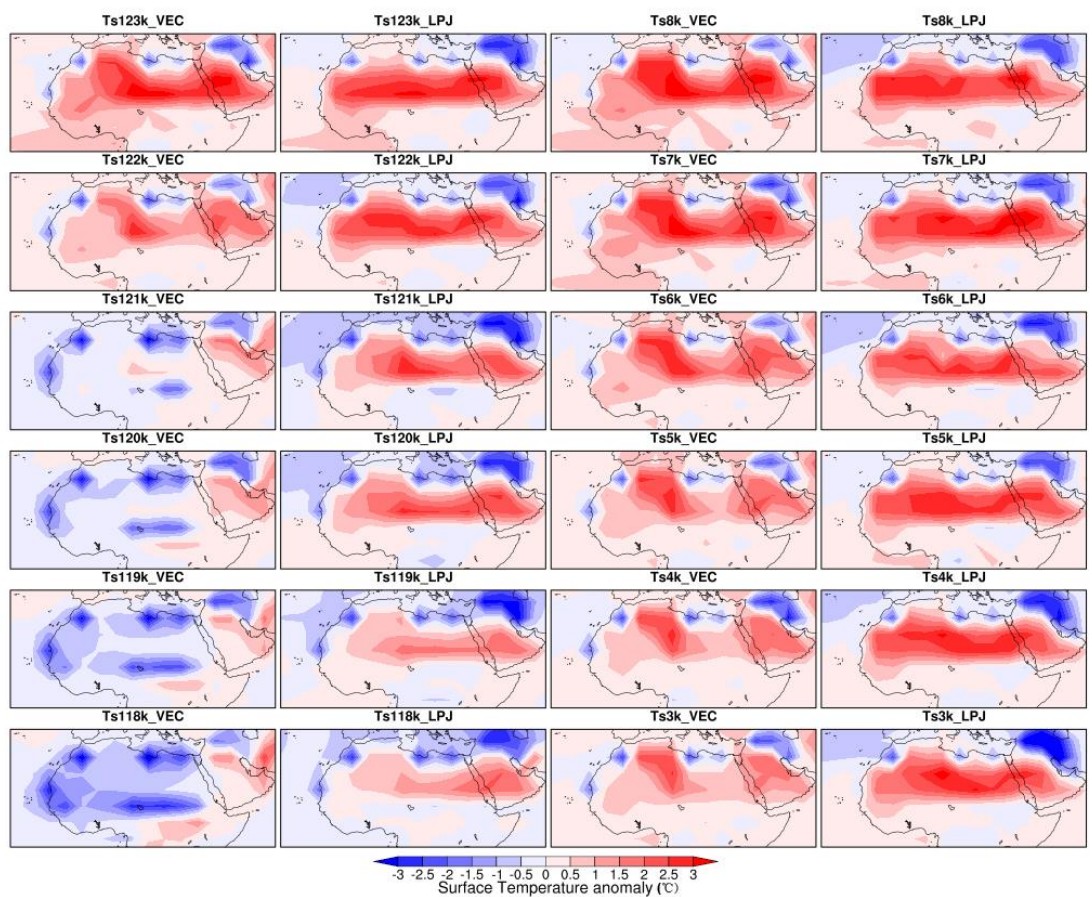


**Figure S3. Surface temperature anomalies during the LIG (left two columns) and Holocene (right two columns), results from experiments with dynamical vegetation compared to those with fixed vegetation from VECODE (left) and LPJ-GUESS (right).**

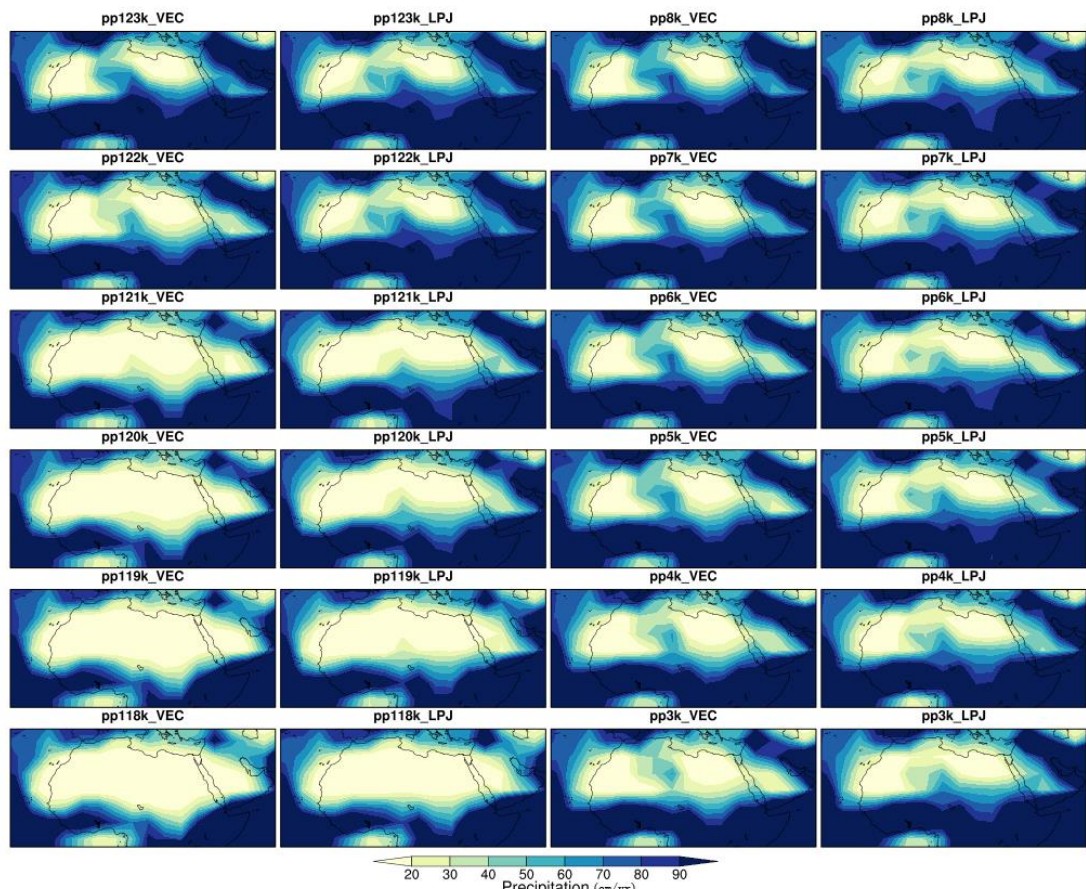


**Figure S4. Precipitation during the LIG (left two columns) and Holocene (right two columns) from VECODE (left) and LPJ-GUESS (right).**

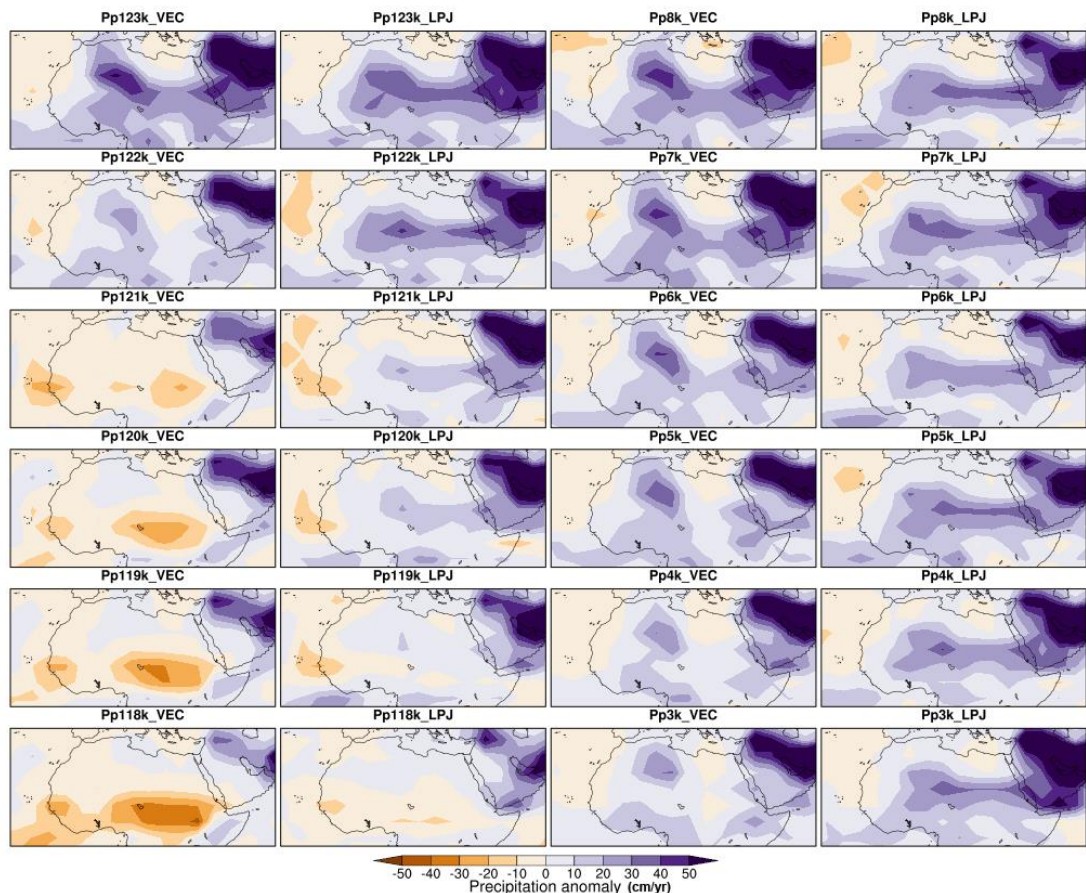

**Figure S5. Precipitation anomalies during the LIG (left two columns) and Holocene (right two columns), results from experiments with dynamical vegetation compared to those with fixed vegetation from VECODE (left) and LPJ-GUESS (right).**

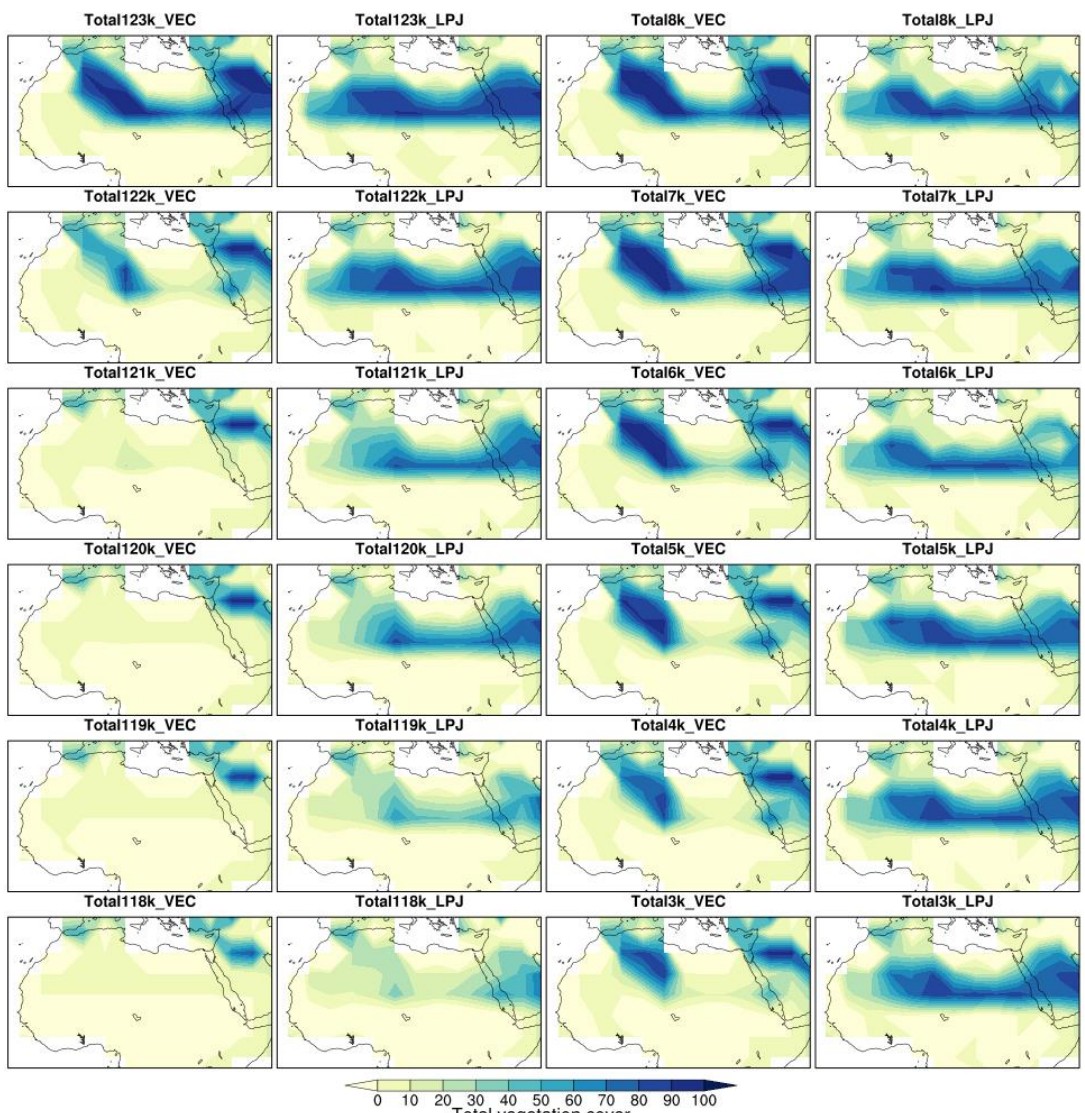

**Figure S6. Total vegetation cover during the LIG (left two columns) and Holocene (right two columns) from VECODE (left) and LPJ-GUESS (right).**
