# Peer review of "Comparison of the Green-to-desert Sahara transitions between the Holocene and the Last Interglacial"

_Climate of the Past, 2022_

## Author Response (AR1)

**Reply to reviewer #1**

*We thank the reviewer for providing very detailed and constructive comments on our manuscript. Below we have replied to all his/her comments (in italic font).*

**Major comments:**

(1) The authors could do a better job of motivating this study. It is not clear why it is necessary to simply compare these two time periods regarding Green Sahara (vegetation-precipitation transition in the Sahara). We understand some studies already modified to simulate Green Sahara reproducing intensification and geographical expansion of the West African monsoon (e.g., Pausata et al. 2016; Hopcroft and Valdes 2021), but many Paleo-modelling still fails to simulate it (Tierney et al. 2017). Therefore, it would be better to have a clear motivation, for example, to obtain clues (regarding vegetation-climate interactions) to modify the Holocene simulation by comparing the two periods. Alternatively, it would be interesting to have a new fact (not known from the data alone) that can be obtained through the comparison.

*REPLY: We thank the reviewer for this good suggestion and for pointing us to additional relevant literature. In the introduction of the revised version, we have now discussed more clearly our motivation for this study and have updated literature references Our main motivation comes from the fact that the abruptness of the termination of a 'Green Sahara' state is still debated and that factors contributing to such a termination are still uncertain. By performing transient climate model experiments on two different interglacials, and by applying two different vegetation models of different complexity, we are able to provide a thorough evaluation of the abruptness and the most important factors. Ideally, similar experiments should be conducted with a range of coupled climate models to extend such an analysis.*

(2) It might be better to describe what is already known in the Introduction section and show more new results in this paper. For example, the spatial and temporal complexity of the termination of the African Humid Period (AHP) have already known (Shanahan et al. 2015; Tierney et al. 2017; Dallmeyer et al. 2020), and data regarding the abruptness of precipitation/vegetation decline is a local- or regional-scale feature (Brovkin and Claussen 2008), not the whole Sahara. In this study, one of the main analyses is to investigate changes in climate (particularly surface temperature and precipitation) and vegetation cover in the whole Sahara or North Africa, including the Sahel, during the LIG and Holocene. However, it is better to analyse the western and eastern parts of the Sahara separately rather than the whole Sahara.

*REPLY: We thank the reviewer for highlighting the important temporal and spatial variations in the termination of the AHP. We are aware of such variations and in fact we already analyse in Figure 4 separately the responses of the western and eastern parts of the Sahara as suggested by the reviewer. We have improved the introduction by highlighting these spatial-temporal aspects more clearly.*

(3) It seems that the data already show that the Green Sahara happened in the two periods, but what else do we know from the data, especially about differences? Finally, is there a reason for no quantitative model-data comparison, in particular, the Holocene has been made at all? We cannot decide whether these simulations are good or bad at all.

*REPLY: We agree with the reviewer that quantitative model-data comparisons can be very powerful ways to evaluate model results. However, we would argue that such a quantitative model-data comparison for the two interglacials would imply a major additional effort that requires a paper of its own. The focus of our paper is on the comparison of different model results for the two interglacials. In our study, we do compare our Holocene results qualitatively with proxy-based evidence and conclude that our model results are consistent with the proxy data. For the LIG, less proxy data are available for a comparison of the nature of the Green Sahara termination. In our revised manuscript, we have briefly discussed (Section 3.4) that evaluation of model results using proxy-based reconstructions is a crucial step.*

Not directly related to this study, but compared to the data (e.g., Hoffman et al. 2017; Capron et al. 2017; Scussolini et al. 2019) how good do the LIG simulation (Li et al. 2020) reproduce global-scale surface temperature and precipitation?

*REPLY: In our previous publication (Li et al. 2020) we discussed the global response to LIG forcings in our model and the agreement with proxy-based evidence. In the present manuscript, we would like to focus on the response in North Africa and the comparison to the Holocene. However, we have added a few lines to the introduction to summarize the performance of the model relative to proxy data.*

**Line comments:**

L.11: "Hand" to "Hans"

*REPLY: Done.*

L.43: "the rate of this transition remains controversial" – The authors would be better to clarify whether these are differences among data or/and discrepancies between data and model.

*REPLY: We have clarified as suggested.*

L.60: Since Liu et al. (2006), many studies have already been implemented on this topic, but why do the authors not mention any recent studies? -- Liu et al. (2006): SC, Claussen et al. (1999): UC, and how about the other recent studies?

REPLY: *We cited the original publications that first reported on these two different types of transitions. We have added more references in the revised manuscript as proposed.*

L. 60: "during the termination of the AHP" to "during the AHP"

*REPLY: We have decided to keep "during the termination of the AHP", as this sentence is about the "rate of vegetation change", so not about the vegetation during the AHP itself.*

L.67 and L.71: I do not understand the mechanism well. Compared to the desert, the vegetated area has lower albedo and absorbs more SW radiation leading to a warmer surface. However, a vegetated surface produces more latent heat and cools the surface (thus warming the atmosphere above). Thus, changes in surface temperature should be determined by the

balance between warming due to increased absorption of SW radiation and cooling due to increased surface latent heat.

*REPLY: We agree with the reviewer and have explained the involved mechanisms in the third paragraph of the introduction.*

L.81: Which specific period is the LIG here? For example, from about 129 ka BP to 120 ka BP.

*REPLY: We thank the reviewer for this point. Indeed, the period with stronger insolation than the Holocene is the early part of the LIG, from about 129 to 120 ka. Since our experiments are focused on the green-desert transition, we start them at 127 ka BP and run until 116 ka BP. This is also consistent with the PMIP4 protocol. We have clarified this in the introduction of the revised version.*

L.82-93: After all, are those simulations quantitatively consistent with the data? Does the LIG data also show a fact of "nonlinear response of the African monsoon to orbital forcing" and "the spatial heterogeneity of the response" as well as the MH data?

*REPLY: We refer to both modelling and reconstruction studies in this paragraph, and both consistently show that there was a "Green Sahara state" also in the LIG. However, as mentioned previously, there exists to our knowledge no clear proxy-based evidence for an abrupt termination of this vegetated state.*

I found a qualitative LIG precipitation data (Scussolini et al. 2019), but do we have any quantitative LIG precipitation data?

*REPLY: To our knowledge, there are no quantitative precipitation data available for the LIG. We have cited the Scussolini et al. (2019) paper in the revised manuscript.*

L.93-95: Since these two sentences are new topics, they could be moved to a new paragraph. Also, is the issue the authors point out here limited to iLOVECLIM, or does it involve other GCMs as well?

*REPLY: We agree with the reviewer and have made a new paragraph in the revised version. This is likely to be applicable to other models as well.*

L.103: Could the authors also use these two vegetation models under the same conditions? In other words, can VECODE also be simulated asynchronously with iLOCECLIM? Is it technically impossible?

*REPLY: Yes, this would be technically possible, but we decided not to apply this in the present study.*

L.109: The scientific significance of the first and second questions is a little unclear to me. Could the authors please elaborate a bit more on why these questions are important?

*REPLY: Question 1 allows us to estimate the impacts of different vegetation components on both interglacials. By answering this question, we know the baseline of vegetation anomalies induced by the differences of vegetation components in iLOVECLIM. We can then answer*

*question 2 by comparing the two interglacials, aiming to understand how orbital forcing and internal feedbacks affect desertification in North Africa. Questions 1 and 2 are important because before we can analyse and compare the feedbacks during the LIG and Holocene, we first have to characterize what happens to the climate and vegetation in the different experiments.*

L.119: The authors can describe a little more about cloudiness, humidity, and precipitation of ECBilt because ECBilt is somewhat different from AGCMs. I understand that ECBilt uses the prescribed/fixed cloud cover based on the modern condition throughout the paleo-simulation.

*REPLY: Yes, ECBilt uses prescribed monthly cloud cover based on observations. This is now added in the revised version.*

L.130: LPJ-GUESS adopts a simple two-layer bucket model (with prescribed percolation rate and water holding capacity), but is VECODE the same/similar structure? If they differ, it would be better to describe the difference.

*REPLY: ECBilt-VECODE uses a one-layer bucket model, so different from what is used in LPJ-GUESS. We have explained this in the revision.*

L.140: LPJ "standard" version (Sitch et al. 2003) has 10 PFTs, but what is the other PFT? Do the authors count bare ground as a PFT?

*REPLY: We have now clarified the 11 PFTs, which are: Boreal needle-leaved evergreen trees, Boreal needle-leaved evergreen shade-intolerant trees, Boreal needle-leaved summer-green trees, Temperate broadleaved summer-green trees, Boreal-temperate broadleaved summer-green trees, Temperate broadleaved evergreen trees, Tropical broadleaved evergreen trees, Tropical broadleaved evergreen shade-intolerant trees, Temperate broadleaved raingreen trees, C3-grass, and C4-grass.*

L.143: It seems that the content here is not a model description, but an experimental design.

*REPLY: This sentence is about the procedure to couple different model components, so in our view it belongs in the Methods section where we discuss the model setup. We therefore prefer to keep it here.*

L.146: The last two sentences in the paragraph are a little unclear for me. Could the authors explain the experimental design for the asynchronously couped version, ECBilt-CLIO_LPJ-GUESS with a chart?

*REPLY: The asynchronous coupling is explained in detail in Li et al. (2020), including an explanatory figure. In the asynchronous coupling procedure, monthly climate inputs from the fully coupled iLOVECLIM model (ECBilt-CLIO-VECODE) are in an initial step fed to LPJ-GUESS, which simulates vegetation distributions. Then, the resulting vegetation distributions are given back to ECBilt, as a fixed vegetation component in iLOVECLIM (ECBilt-CLIO_LPJ-GUESS) during the next round of climate simulation. A new climatology from this integration is simulated and used subsequently off-line as climate forcing by LPJ-GUESS to produce a new global vegetation distribution that is subsequently used as a boundary condition by iLOVECLIM, and so on. We have clarified this in the fourth paragraph of Section 2.1.*

L.154: 1850 AD, not 850 AD (for prescribed pre-industrial condition) typo(?)

*REPLY: It is in fact 850 AD. This year is used because the anthropogenic disturbance of the natural vegetation was still relatively modest.*

L.157: The description of LBM should be moved before the paragraph on each of the three-model configuration.

*REPLY: We do not readily see why this part should be moved, as it is valid for all the three model configurations discussed in the sections just before this part. We therefore decided to keep it here.*

L.162: LPJ-GUESS has 11 PFTs, but did those PFTs simply convert into 3 types (trees, grasses, and desert) for the LBM?

*REPLY: Yes, such a conversion takes place. We have explained the conversion in our earlier publication Li et al. (2019a). This is clarified in the revised manuscript.*

L.167: This paragraph is a bit confusing. Does it mean that soil hydrology calculated in LPJ-GUESS does not directly affect ECBilt, but has some indirectly influence through vegetation type?

*REPLY: Yes, the procedure is as described by the reviewer. We explain this on lines 171-173. We have revised the text for clarification.*

L.185: At each time-slice simulation (HOL_LPJ, LIG_LPJ), how many model years did the authors run the model and how many years of the output were used in the analyses?

*REPLY: LPJ was run for 1000 years per time slice, of which we used the last 30 yrs. This is clarified in the revised manuscript.*

L.213: What is the range of the target area (latitudes and longitudes) for North Africa or the Sahara here?

*REPLY: We took 10W-35E, and 15N-30N as limits for our analysis, as in Li et al. (2020). We have clarified this in the figure caption of Figure 1 in the revised version.*

L.223-225: How about the recent (CMIP6/PMIP4) simulations about? Comparison with past simulations is important, but comparison with recent simulations as well as data is also important.

*REPLY: We checked the recent CMIP6/PMIP4 simulations for the mid-Holocene and the Last Interglacial (Williams et al., 2020) and updated this part with most recent LIG simulations in the revised version.*

L.225: Why is the LIG_FIX temperature trend positive?

*REPLY: In LIG_FIX, the vegetation is fixed to desert in the entire experiment, so there are no changes in albedo as in the experiments with dynamical vegetation. Even without the albedo effect, the precipitation in Northern Africa was still significantly higher in the early part of*

*the interglacial due to the enhanced summer monsoon, forced by elevated insolation values. This high precipitation resulted in relatively humid soils and enhanced evaporation, leading to evaporative cooling in the first part of the LIG relative to the end of the LIG_FIX experiment. This created the positive temperature trend that is also seen in other LIG experiments without dynamical vegetation (e.g., Bakker et al. 2014). We have clarified this in the revised paper.*

L.228: Because LIG_LPJ does not show large changes in surface temperature in North Africa, are changes in surface temperature and desertification (vegetation cover) less relevant in this simulation?

*REPLY: The surface temperature changes in LIG_LPJ are less expressed than in LIG_VEC, so they seem less relevant for the desertification. However, the response in LPJ_LIG is more regional, with a relatively strong response in the western part of the Sahara (see Figure 3).*

L.235: Fischer and Jungclaus (2010) analysed time-slice simulations, not transient ones. So that may not be an appropriate reference here. Moreover, according to Brovkin and Claussen (2008), which is also cited in this paper, Francus et al. (2013) may not be an appropriate reference either, because the individual data represent local responses and are not representative of the whole North Africa.

*REPLY: We thank the reviewer for pointing this out. We have adjusted the referencing accordingly.*

L.235~237: Figure 1f shows that magnitude of precipitation decline in HOL_LPJ is similar to one in HOL_FIX, and this sentence may not be appropriate.

*REPLY: We agree and have corrected this in the revision.*

L.245: Change Fig. 2f to Fig. 1f or Fig. 2b(?) Anyway, we cannot consider "the simulated vegetation distribution and spatial divergence in North Africa" from this figure, I think.

*REPLY: We thank the reviewer for pointing this out. We have now referred to Figure 3 which shows spatial responses in vegetation.*

L.253: It seems that surface temperature trend in HOL_LPJ is similar to one in HOL_FIX.

*REPLY: Acknowledged. We have added "similar to HOL_FIX".*

L.257: Can we check the ratio of trees to grasses in North Africa? Vegetation-induced changes in albedo and surface evaporation may also depend on the surface conditions between trees and grasses.

*REPLY: Yes, we can technically calculate the ratio of trees to grasses in North Africa. In fact, on the one hand, tree-cover was very limited in North Africa (the maximum tree cover of 16% at 127 ka BP was simulated in LIG-VEC, while the tree cover was less than 5% in the meantime in LIG_LPJ) and the changes of grass and desert cover are in phase; on the other hand, we can see declines in both tree and grass cover during the two interglacials, and neither trees nor grasses exist in North Africa after desertification. Therefore, we considered*

*vegetation-induced changes through total vegetation cover rather than tree-induced and grass-induced separately.*

L.262: What is the reason for the sharp decline in vegetation cover, especially from 123ka o 121 ka in the LIG_VEC simulation? Moreover, why is that trend not seen in HOL_VEC?

*REPLY: The sharp decline in vegetation in LIG_VEC is simultaneous with a strong reduction in precipitation (Fig. 1b), but also with stronger cooling (Fig. 1c), showing that feedbacks between vegetation and climate are behind the enhanced desertification. We have clarified this in the revised text (first paragraph of Section 3.2.1).*

L.263: Can the authors check how much the ratio of trees to grass in North Africa varies from model to model? Looking at the vegetation area fraction anomalies (Fig. 3), there may be considerable differences between the two models in terms of the proportion of trees and grasses.

*REPLY: Yes, in both LIG_LPJ and HOL_LPJ tree-cover remains less than 5%, and the main declines of vegetation cover were contributed by grass. Compared to these LPJ-experiments, tree cover was somewhat higher in both LIG-VEC and HOL-VEC during the early periods of both interglacials although tree-cover was also very limited. During the desertification, both trees and grass decline until they disappear.*

About the different vegetation diversity between the two models, unlike Claussen et al. (2013) VECODE and LPJ-GUESS are completely different process-based DGVMs, and there must be many differences besides diversity.

*REPLY: We agree with the reviewer. More general, it is related to a difference in model complexity. In an earlier paper (Li et al. 2019a), we have tested the impacts of DGVMs with different complexity on vegetation simulations under different climate conditions. Based on our conclusion, the complexity of VECODE and LPJ-GUESS affects vegetation simulations mainly through diversity when the atmospheric $CO_2$ level is around pre-industrial level (280 ppmv), while the difference in complexity affects vegetation simulations mainly through ecophysiological processes when the atmospheric $CO_2$ level is largely different from PI level. We have clarified this in the text (first paragraph of Section 3.2.1).*

L.266: "Claussen et al. 2013", not "Claussen, 2009" I think.

*REPLY: We have revised as suggested.*

L.272 and L.347: Yu et al. (2017) proposed the observed positive vegetation feedback on precipitation in the Sahel (not the Sahara) by a moisture recycling mechanism rather than the classic albedo-based mechanism. Messori et al. (2018) have a similar idea for the Holocene Green Sahara. Does this concept apply to the authors' experiments (the Sahara in the LIG and Holocene)?

*REPLY: There are three main biogeophysical feedbacks in this process, which are the positive vegetation-albedo-temperature feedback, the negative vegetation- evaporation-precipitation feedback (Liu et al., 2007; Notaro et al., 2008; Wang et al., 2008) and the positive vegetation-evaporation-precipitation feedback (Yu et al., 2017; Messori et al., 2018). Both the negative and positive feedback to precipitation are not seen in our experiments, which could be related*

*to the fixed surface albedo of bare soil that is not a function of water content in both our dynamical vegetation experiments. This could lead to an overestimation of the positive vegetation-albedo feedback in our simulations, and we have discussed this in the manuscript.*

Fig.3: PFT fraction anomalies for LIG simulations (in particular 120k ~ 118k) show shrinking vegetated areas extend much southerly, but why does that feature not happen in Holocene simulations?

*REPLY: This difference between the later parts of the interglacials is related to the difference in insolation forcing. As can be seen in Figures 1a and 1e, the summer insolation decreases more strongly in the LIG, with a value well below that of the Holocene between 120 and 118 ka. We have explained this more clearly in the 3ʳᵈ paragraph of Section 3.3..*

L.278: What do the authors think caused the decline with large error bars at 5 ka and 4 ka in the HOL_LPJ simulation? Moreover, why does that feature not catch in the LIG_LPJ simulation?

*REPLY: In the HOL_LPJ snapshot experiments for 5 and 4 ka there was a strong increase in the interannual variability in the simulation of PFT cover. This is likely related to one or more PFTs being very close to their climatic limit, such that relatively small variations in precipitation cause large shifts between the area covered by the involved PFT. In this case, also bare ground was involved. In the LIG_LPJ experiments the climatic input was different, meaning that the PFT response was also different. We have added a comment on this feature in the first paragraph of Section 3.2.2.*

L.279~284: The authors should describe the spatial heterogeneity in the Introduction section, not here because this is a known fact. Furthermore, based on this, from the beginning, the region should be divided into East and West for analysis, I think.

*REPLY: We agree with the reviewer that this spatial heterogeneity should be treated in the introduction. We have mentioned this now in the revised version. However, we would argue that it is also useful for the reader to discuss it briefly here in 3.2.2, as it relates to the model results discussed in this section.*

L.284: As mentioned before, the authors can confirm this by making VECODE asynchronous with iLOVECLIM.

*REPLY: We agree that it would be interesting to perform experiments with asynchronous coupling to VECODE as well to compare with the results from the synchronous coupling and see what the effect of the different forms of coupling is. However, unfortunately performing such additional experiments is not feasible for us. Besides, since LPJ and VECODE are very different models, it is questionable if any inferences for synchronous vs asynchronous coupling with VECODE are also valid for LPJ.*

L.292: What are the grid points for both western and eastern North Africa, respectively?

*REPLY: We divided the area in two, 10W-10E, and 10E-35E, both with latitudinal limits at 15N and 30N. We have clarified this in the caption of Figure 4.*

L.298: It's hard to see the differences between West and East Sahara from Figure 4. Moreover, what exactly is "A spatial and temporal complexity of the termination of the AHP"?

*REPLY: In our opinion, the results for West and East Sahara are similar, but still differences can be clearly seen. For instance, precipitation and vegetation cover was clearly lower in the early Holocene in the Eastern Sahara. Spatial complexity refers to this difference between east and west. In relation to the lower precipitation in the east, the AHP ends earlier here than in the west. The temporal complexity refers to this difference in timing. We have rephrased the text to clarify this (2nd paragraph of Section 3.2.2).*

L.302: About the sentence "the magnitude of our vegetation decline is much weaker than in their study", which study/value matches the available data?

*REPLY: The mentioned sentence refers to Liu et al. (2007).We added the value of vegetation declines from Liu et al. (2007) in the revised version.*

L 302: Is "the differences in model complexity" simply about the vegetation models between Liu et al. (2007) and this study?

*REPLY: Yes, that was meant here.*

L.321-323: Does any data also support the changes in climate and vegetation in the LIG are stronger than ones in the Holocene?

*REPLY: To our knowledge, data on the LIG are too sparsely to be conclusive on this point.*

L.328: Is around 125 ka BP and around 8.5 ka BP each peak of insolation at 20N for the LIG and Holocene respectively?

*REPLY: During the LIG, the summer insolation at 20N peaks at 125ka as can be seen in Figure 1, but during the Holocene the peak was at 10 ka, so a bit earlier than 8.5 ka.*

L.338: Did Shanahan et al. (2015) discuss the vegetation cover and vegetation-albedo feedbacks using TraCE-21 simulation?

*REPLY: Shanahan et al. (2015) compared proxy data with the results from the TraCE simulations (see their Fig. 1 and Supplementary Figures S8 for example).*

L.341-344: Does any data also support the changes in climate and vegetation in the LIG are stronger than ones in the Holocene? Or, will the results of the LIG simulation help to improve Holocene simulation?

REPLY: *As mentioned before, to our knowledge there are not sufficient data on the LIG to be conclusive on this point.*

L.361: In this study, "the fractional surface albedo of trees, grassland, and desert are seasonally fixed". Could this setting also be relevant?

*REPLY: Yes, we agree with the reviewer that this could play a role. We have added this in the revised version.*

L.380: It seems that the section 3.4 is not what the authors found out through comparison between the LIG and Holocene simulations. How about discussing at least one issue that arose through comparison?

*REPLY: We are not sure what the reviewer means here. Is the suggestion to discuss an uncertainty issue that is related to the comparison? We could for instance add that, compared to the Holocene, it is difficult to evaluate the LIG experiments because of the sparsity of appropriate proxy data.*

**Reply to reviewer # 2 (Qiong Zhang)**

*We thank the reviewer for providing constructive comments on our manuscript. Below we have replied to all her comments (in italic font).*

1. There are more proxy evidence on the green Sahara during Holocene, while less reporting on a green Sahara and the abrupt transition from green to desert during LIG from the proxy aspect. Line 85 cited three papers on such evidence, it would be good to provide more detailed information on their findings , such as what kind of proxy, the location and what do the data imply.

*REPLY: This is a good point. We have provided in the revised manuscript (4$^{th}$ paragraph of the Introduction) the information proposed by the reviewer, and in addition references to additional publications.*

2. In model description, more information on the physics of the atmospheric model ECBilt should be provided, since the manuscript mainly discuss the changes in precipitation. When coupled to the LPJ-GUESS module, it also uses could cover as one climate input, the description on relevant physics such as the cloud and convection of the model would be helpful to understand the simulated precipitation and climate and how they further influence the vegetation simulation. For the model resolution T21, would be good to provide the grid distance in kilo-meters as a reference to paleoclimatologist who are not familiar with the spectral grid.

*REPLY: Much of this information was provided in our previous paper Li et al. (2020), but we agree with the referee that more information on ECBilt and the coupling to LPJ would be useful for the reader. We have provided this information in the revised manuscript (1$^{st}$ paragraph of Section 2.1).*

3. For the coupling to vegetation model, climate input for LPJ-GUESS is the monthly mean, while VECODE uses annual mean temperature, precipitation and GDD0 (Line 132). We know that the changes in seasonality due to orbital forcing are the major cause for changes in African monsoon, the authors need to comment the potential effect by using annual coupling with dynamic vegetation model VECODE.

*REPLY: We thank the reviewer for this suggestion. We have added a few lines on this point in the revised manuscript (3$^{rd}$ paragraph of Section 2.1). Potentially, using annual climate forcing for VECODE could imply an underestimation of the impact of orbital forcing on the vegetation development in North Africa. However, please note that VECODE receives some seasonal information through the GDD0 (growing degree days above 0°C).*

4. In line 150, it is mentioned that pre-industrial vegetation from the CMIP LUH2 dataset is upscaled and used as the prescribed vegetation in PI_FIX, it would be helpful to show this pre-industrial vegetation map, and mention what information from the vegetation map is read by the ECBilt (albedo, evaporation?). Also good to compare this vegetation pattern with the simulated ones in PI-VEC and PI_LPJ, this would help to image and understand the changes in vegetation cover in Fig3.

*REPLY: We have included a map showing the PI vegetation used in the experiments with fixed vegetation. The vegetation map is used to infer land surface albedo and the maximum water volume in the bucket model.*

5. When present the area averaged vegetation cover and climate parameters for north Africa in Fig1 and Fig2, should mention the domain for the average. If they are averaged over the entire region north of equtor showed in fig3, it includes the equatorial African region and African monsoon region, which are known as wet and vegetated even today when Sahara is desert. One might wonder what happens in these regions from 121K when vegetation cover is close to zero in Fig2, even equatorial and monsoon region became desert?

*REPLY: This is a good point that was also commented on by reviewer 1. The used domain has as limits 10W-35E and 15-30N, so covers the present-day Sahara region and part of the Sahel. The modern African monsoon region is not included. We have clarified this in the revised manuscript.*

6. I understand the variables showed in Fig1 are annual mean. Following the given insolation in summer, one might wonder why the annual mean temperature has warming trend when the summer insolation has the decreasing trend in the case of no vegetation feedback. Even though the authors mentioned in Line 225 that this warming trend is also seen in other simulations, would be good to explain what cause the warming trend, it can not be due to the GHG forcing since GHG is fixed.

REPLY: *This point was also raised by reviewer 1. In LIG_FIX, the vegetation is fixed to desert in the entire experiment, so there are no changes in albedo as in the experiments with dynamical vegetation. Even without the albedo effect, the precipitation in Northern Africa was still significantly higher in the early part of the interglacial due to the enhanced summer monsoon, forced by elevated insolation values. This enhanced precipitation resulted in relatively humid soils and enhanced evaporation, leading to evaporative cooling in the first part of the LIG relative to the end of the LIG_FIX experiment. This created the positive temperature trend that is also seen in other LIG experiments without dynamical vegetation (e.g., Bakker et al. 2014). We have clarified this in the revised paper (2nd paragraph of Section 3.1.1).*

7. Fig3 showed an interesting anomaly pattern in vegetation cover, I am curious if such a pattern in supported by the proxy data or other model simulations. the authors mentioned in line 297-298 about the spatial complexity in two references, would be nice to provide more information from these findings.

*REPLY: As suggested by the reviewer, we provided more detailed information from these findings in the revised version.*

8. The manuscript focuses on the interaction between the climate and vegetation, it would be helpful to show the spatial anomaly pattern of precipitation, temperature and soil moisture, in order to understand how the climate anomaly pattern affect the vegetation anomaly pattern in Fig3.

*REPLY: As proposed by the reviewer, we provided some maps showing the spatial distribution of climate variables in the supplementary information as background information (Supplementary figures S2 to S6).*

9. In Fig.4 the simulation for HOL_VEC did not show the full simulation period and stopped at 2000, and HOL_LPJ stopped at 3000 yr BP, what is the reason? Something strange happened in the late periods?

*REPLY: Nothing strange happened, we just focused on the results showing the transition, which ended before 3 ka BP.*

Fig 6, can you explain why the model simulate more precipitation during Holocene than during LIG, e.g., when insolation is the same, despite with fixed or coupled vegetation. According to Fig1, insolation declined below 460 W/m2 after 121 K during LIG and after 3 K during Holocene, in Fig2 almost no vegetation after 119 K but some 5-10% remain after 3K in coupled VECODE, please comment on what cause these differences. Please also comment on the possible reason for an accelerated increase in precipitation during LIG when insolation greater than 480 W/m2

*REPLY: Differences in ocean surface temperature are likely to play a role here. The monsoonal precipitation is depending on the thermal contrast between the ocean and continent. The ocean was slightly warmer in the LIG than in the Holocene, which could partly explain a stronger precipitation in the Holocene with the same summer insolation. In addition, vegetation-climate feedbacks play a role. For example, as can be seen in Figure 1, the insolation was similar at 122 ka and 8 ka. However, Figure 3 shows that the vegetation cover is more extensive at 8 ka than at 122 k, which enforces the precipitation through the vegetation climate feedbacks. We have included a paragraph on this clarification in the revised version of the manuscript (5th paragraph of Section 3.3).*

---

## Author Response (AR2)

**Reply to the editor and reviewers**

Editor (Martin Claussen)

Dear authors,
Thank you very much for the careful revision. The reviewers suggest publication 'as is' and 'subject to technical correction'. I would like you to check Figure S6. Is this really the total vegetation cover? If it would be, then there were almost no vegetation in Subsaharan Africa.

*Reply: We have corrected Figure S6. Thank you for pointing this out*

Anonymous Referee #1
I am very satisfied with revised manuscript, which adequately answers many of the questions I previously asked. However, before publication, the authors still need to modify two minor points, I think.

The first point is in Section 3.4, uncertainties. What uncertainty you are discussing is a bit unclear to me. Do you mean uncertainty of the timing of the termination of the AHP between west and east Africa? Or uncertainty of the magnitude of positive-negative vegetation feedback between west and east Africa? Or is it something else?

*Reply: This paragraph is about the uncertainty in spatial vegetation response. Our climate model has a low spatial resolution and does not capture subtle spatial differences in vegetation-climate interactions. This implies that any abrupt change in vegetation that occurs at a regional scale will not be represented in the model, effectively resulting in a smoother response. We have revised the text to clarify this point.*

The second point is regarding English writing. This manuscript has good enough scientific content but some English issues. I am not a native English speaker, but I will just point out the problems in Introduction. If you think my point is correct, please correct it. Then, if necessary, please check the other sections.

Line 35. 'present' to 'the present'
Line 37. 'This period is therefore referred to as the African Humid Period (AHP), featured shrub- and grass-covered land surface in Northern Africa, where there is desert today (…).'
Line 46. 'a dramatic desertification' to 'dramatic desertification'
Line 47. 'a slower desertification' to 'slower desertification'
Line 55. 'There were several studies (e.g., …) that investigated' to 'Several studies (e.g., ...) investigated'
Line 62. '… controversial and only …' to ''… controversial, and only …''
Line 63. 'the desertification/vegetation-transition is' to 'the desertification/vegetation transition is''
Line 67 and 68. 'so called' to 'so-called'
Line 71. 'collapse as a result of' to 'collapse resulting from'
Line 83. 'surface.' to 'surfaces.'
Line 84. 'the evaporation' to 'evaporation'
Line 89. 'different type' to 'different types'
Line 92. 'vegetated surface' to 'a vegetated surface', 'the vegetated surface', or 'vegetated surfaces'
Line 98. 'in the Holocene (), but with …' to 'in the Holocene () but with …'
Line 105. 'to present day' to 'to the present day'
Line 106. 'as a result of vegetation feedbacks' to 'due to vegetation feedbacks'

Line 106. 'doubling of precipitation' to 'doubling precipitation'
Line 107. 'we have performed' to 'we performed'
Line 108. 'a clear warming' to 'clear warming'
Line 121. '(iLOVECLIM) including' to '(iLOVECLIM), including'
Line 124. 'model-dependence' to 'model dependence'
Line 125. 'model-dependency' to 'model dependency'
Line 126. 'by performing … interglacials, and by applying …' to 'by performing … interglacials and by applying …' or 'by performing … interglacials and applying …'
Line 131. 'between both interglacial periods' to 'between interglacial periods'

*Reply: We thank the reviewer for suggesting all these textual improvements. We revised the text accordingly.*